# Structure-Centric Graph Foundation Model via Geometric Bases

Xiaodong He [* 1]   Haolan He [* 1]   Ruiyi Fang [2]   Ming Sun [1]   Zhao Kang [† 1]

## Abstract

Graph foundation models (GFMs) seek transferable representations across graph domains but are limited by structural heterogeneity and incompatible node feature spaces. We propose Structure-Centric Graph Foundation Models (SCGFM), which treat graph topology as the primary source of transferable knowledge. Modeling graphs as metric measure spaces, SCGFM introduces learnable geometric bases that define a shared structural coordinate system. Graphs are aligned to these bases via Gromov–Wasserstein distances, yielding structure-aligned latent representations that accommodate heterogeneous graph topologies. To address feature incompatibility, SCGFM employs a structure-aware feature re-encoding mechanism that unifies node representations without assuming a fixed feature dimensionality or requiring dataset-specific preprocessing. Experiments on graph- and node-level tasks demonstrate strong in-domain and cross-domain generalization, outperforming existing GFM approaches.

## 1. Introduction

Foundation models have transformed natural language processing (Devlin et al., 2019; Brown et al., 2020) and computer vision (Dosovitskiy et al., 2021; He et al., 2022) through large-scale pretraining and strong cross-domain generalization. These successes have motivated growing interest in Graph Foundation Models (GFMs), which aim to extend the same paradigm to graph-structured data. Existing approaches largely follow two directions: (1) augmenting graph models with large language models via prompting or adapters to inject linguistic priors (Ye et al., 2024; Wang et al., 2023; Zhao et al., 2023), and (2) pretraining graph

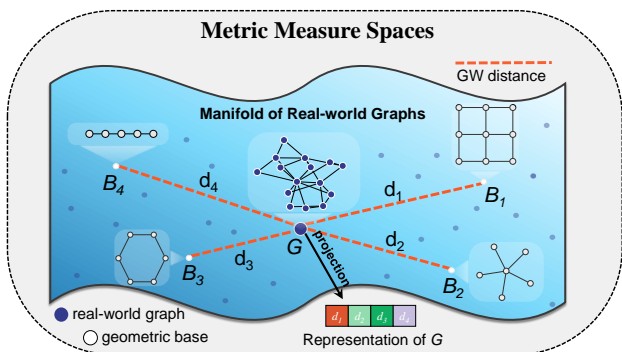

*Figure 1.* **Geometric perspective of SCGFM.** We view graph representation learning as a triangulation process in the space of metric measure spaces. Real-world graphs are assumed to lie on a low-dimensional manifold (blue region). Given an input graph $G$, its structural discrepancies to a shared set of geometric bases $\{B_k\}$ are measured using the Gromov–Wasserstein (GW) distance. These distances define a coordinate representation of $G$ as $\mathbf{q} = [d_1, d_2, \ldots]$, yielding a unified embedding that aligns heterogeneous graphs across domains.

neural networks on large graph corpora using contrastive or generative objectives (Wang et al., 2025c; Yuan et al., 2025). Despite promising progress, current GFMs still struggle to generalize reliably across graph datasets drawn from disparate domains and distributions.

A central challenge is the lack of a shared geometric reference for graphs (Liu et al., 2025). Unlike text or images, graphs are relational objects defined only up to isomorphism and vary substantially across domains in topology, scale, and induced metrics. Most existing GFM frameworks address this heterogeneity by enforcing a fixed node feature dimensionality (Hu et al., 2020; Rampášek et al., 2022), typically via dataset-specific projections, padding, or dimensionality reduction, often implemented through feature adapters (Yuan et al., 2025; Shen & Kang, 2025). While effective in practice, these strategies align feature spaces without aligning graph structure, causing learned representations to poorly reflect intrinsic graph similarity and limiting cross-domain transfer (Wang et al., 2024).

Recent work has explored graph tokenization (Chen et al., 2025; Wang et al., 2025a) as an alternative, discretizing graphs into symbolic tokens analogous to words in language models. However, this paradigm is fundamentally

---
[*]Equal contribution   [†]Corresponding authors.   [1]University of Electronic Science and Technology of China, Chengdu, Sichuan Province, China [2]Western University, Canada. Correspondence to: Xiaodong He <hexiaodong24@126.com>, Zhao Kang <zkang@uestc.edu.cn>.

*Proceedings of the $43^{rd}$ International Conference on Machine Learning*, Seoul, South Korea. PMLR 306, 2026. Copyright 2026 by the author(s).

misaligned with the geometric nature of graphs. Graphs are non-Euclidean objects and permutation-invariant (Bruna et al., 2013; Fang et al., 2025a; Liang et al., 2025), whereas tokenization typically imposes artificial orderings on inherently unordered structures (You et al., 2018). As a result, token-based representations often violate permutation invariance and fail to capture intrinsic graph geometry, leading to limited robustness and generalization (Bronstein et al., 2021; Jin et al., 2018).

In this work, we adopt a geometric perspective and model graphs as **metric measure spaces** (mm-spaces), in which structure is defined independently of node identities and feature semantics. Motivated by results from metric geometry, we posit that real-world graphs lie within a structured and bounded subset of the space of mm-spaces. Under this assumption, graphs can be represented by their Gromov–Wasserstein (GW) distances (Mémoli, 2011) to a finite set of canonical geometric patterns. This distance-based projection induces a shared, continuous representation space that naturally accommodates heterogeneous graph structures (Figure 1).

Building on this insight, we propose **Structure-Centric Graph Foundation Models (SCGFM)** based on learnable geometric bases. Each base encodes a canonical graph structure with a fixed number of nodes and trainable edge weights, collectively forming a shared **structural coordinate system**. Input graphs are aligned to this system via structure-preserving GW mappings, yielding unified latent representations that explicitly encode graph topology. Unlike prototype or dictionary learning methods that operate primarily in feature space (Zeng et al., 2023; Vincent-Cuaz et al., 2021), our geometric bases are optimized to capture intrinsic structural variability across graphs.

Beyond structural heterogeneity, incompatibility among node feature spaces presents an additional (Wang et al., 2025b; Shen et al., 2024; Li et al., 2024). Rather than enforcing a fixed feature dimensionality, SCGFM leverages structural alignment to induce feature unification: node features are aggregated and re-encoded through their correspondence to geometric bases. This produces unified feature representations anchored to structure, while remaining agnostic to the original feature dimensionality and semantics.

Together, these components yield a unified and scalable GFM that is both structure-centric and feature-flexible. SCGFM[1] can be pretrained on graphs from diverse domains and transferred across datasets without architectural modification or dataset-specific preprocessing, offering a principled alternative to graph tokenization and domain-dependent GFMs. Our contributions are summarized as follows:

---

[1]Code is available at: https://github.com/Xd-He/SCGFM

- We propose a structure-centric graph foundation model that enables transfer across heterogeneous graph datasets without dataset-specific architectures or feature preprocessing.

- We show that the learned geometric bases induce a geometry-aligned latent space whose distances correlate strongly with intrinsic Gromov–Wasserstein graph geometry.

- Extensive few-shot evaluations on graph- and node-level tasks demonstrate strong in-domain and cross-domain generalization.

## 2. Related Work

Graph representation learning has evolved from self-supervised pre-training to GFMs designed for broad transferability across datasets and tasks (Liu et al., 2023; Xie et al., 2023; Fang et al., 2026; Liu et al., 2025; Pan & Kang, 2023). However, *cross-domain few-shot* transfer remains challenging due to (i) substantial *topological shifts* and (ii) *heterogeneous* or missing node features, which undermine feature-centric approaches.

### 2.1. Self-Supervised Graph Pre-training

Early methods focused on contrastive views (e.g., DGI (Velickovic et al., 2019), GraphCL (You et al., 2020)), while later approaches minimized handcrafted augmentations (GraphACL (Xiao et al., 2023)) or adopted scalable masked modeling (GraphMAE2 (Hou et al., 2023), S2GAE (Tan et al., 2023)).

Recent works improve generalization via unified objectives (GALE (Wang et al., 2025e)) and structure-aware curriculum masking (CurMGAE (Li et al., 2025) and SAGA (Fang et al., 2025b)). While effective in-domain, these methods typically assume compatible feature spaces within a shared embedding domain. Consequently, they often struggle with topology-driven shifts or inconsistent features under few-shot settings, limiting cross-dataset transferability.

### 2.2. GFMs Across Domains and Tasks

GFMs unify learning via shared interfaces and large-scale pre-training (Liu et al., 2025; Wang et al., 2025f). Interface-centric designs align tasks using language or structured prompts (OFA (Liu et al., 2024), GIT (Wang et al., 2025g), UniGraph (He et al., 2025), SIGOOD (Wang et al., 2026)). Recent research emphasizes cross-domain robustness: MDGFM (Wang et al., 2025c) and SAMGPT (Yu et al., 2025) focus on topology and structure alignment, while BRIDGE (Yuan et al., 2025) explores selective knowledge assembly. Retrieval-augmented models like RAG4GFM

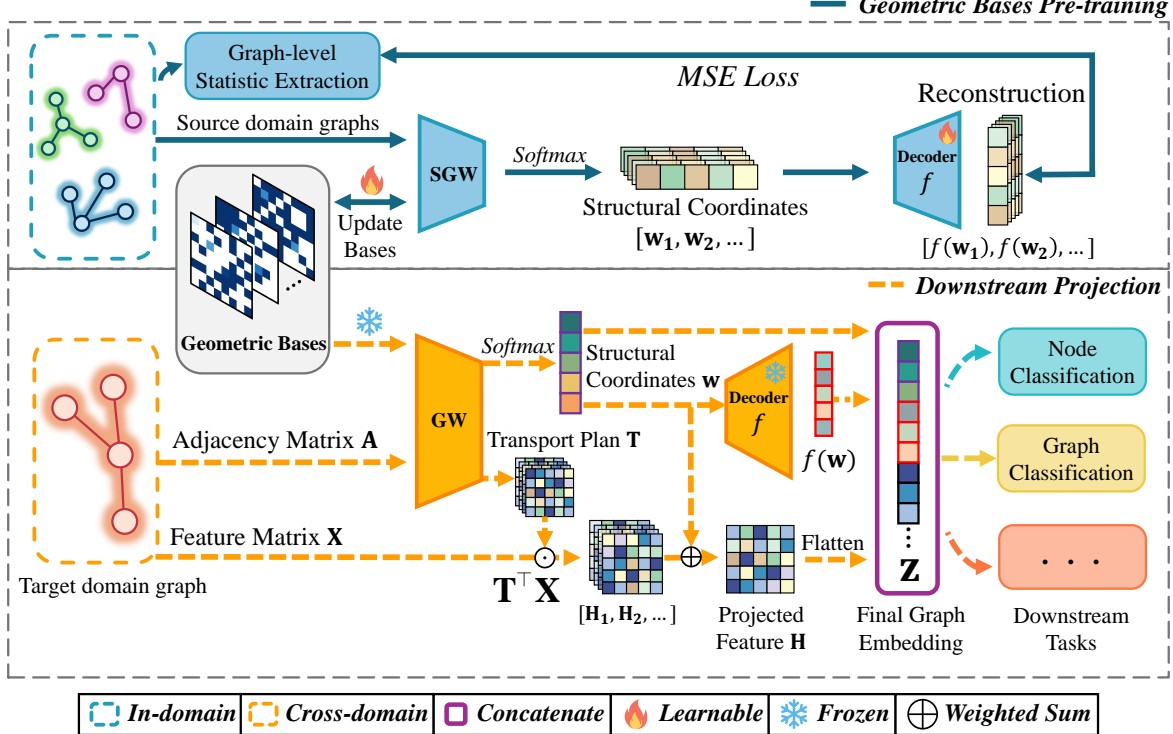

*Figure 2.* **Overall framework of SCGFM.** In **Geometric Bases Pre-training**, trainable geometric bases are optimized via the Sliced Gromov-Wasserstein (SGW) distance to map each source-domain graph to a structural coordinate. In **Downstream Projection**, a target graph is matched to the bases to obtain a structural coordinate vector and a GW transport plan for projecting the feature matrix. The final graph embedding is the concatenation of the structural coordinate $\mathbf{w}$, decoder output $f(\mathbf{w})$, and projected feature $\mathbf{H}$ for downstream tasks.

(Wang et al., 2025d) and RAG-GFM (Yuan et al., 2026) further enhance inference reliability. However, most GFMs still rely on feature alignment or prompts that fail when target graphs exhibit strong topological shifts and unreliable features.

### 2.3. GW-based graph methods.

Several recent works have applied optimal transport to graph-structured data. Vayer et al. (Titouan et al., 2019) introduced Fused Gromov–Wasserstein (FGW) to jointly compare node features and graph structures, while Chen et al. (Chen et al., 2023) employed a GW geometric perspective for spectrum-preserving graph coarsening. These studies highlight the usefulness of GW-type distances for graph comparison and representation learning. In contrast, SCGFM learns a set of geometric bases that define a shared structural coordinate system.

## 3. The Proposed Method

We propose **SCGFM** that learns a finite set of geometric bases to represent arbitrary graphs within a unified metric latent space. It consists of two stages: (1) **Geometric Bases Pre-training**, and (2) **Downstream Projection**. An

overview of the framework is shown in Figure 2.

### 3.1. Geometric Motivation: Finite Covering of Graph Space

We regard the set of all finite graphs, endowed with normalized metrics and measures, inducing elements in a common mm-space, and denote by $\mathcal{X}$ the resulting space equipped with the GW distance $d_{GW}$. Under this view, graph representation learning amounts to learning over the space $(\mathcal{X}, d_{GW})$, where each graph is represented by its relational geometry rather than explicit node correspondences.

**Assumption 3.1 (Total Boundedness).** We assume real-world graphs lie in a totally bounded subset $\mathcal{K} \subset \mathcal{X}$. By Gromov's compactness theorem for mm-spaces (Gromov, 1999):

- Total boundedness is equivalent to the existence of finite $\epsilon-$covers under the GW metric.

- Thus, for any tolerance $\epsilon > 0$, there exist prototypes $\{C_1, \ldots, C_K\}$ such that each graph $G \in \mathcal{K}$ satisfies

$$d_{GW}(G, C_k) < \epsilon \quad \text{for some } k. \tag{1}$$

To instantiate this theoretical guarantee, we propose learning a discrete set of geometric bases that approximates the optimal $\epsilon$-cover directly from data. Unlike task-specific templates, these learnable prototypes induce a shared geometric coordinate system, in which any graph can be represented through its transport discrepancies to the bases. We next specify the parameterization of these bases and show how it ensures that each prototype constitutes a valid mm-space.

### 3.2. Geometric Bases Pre-training

Each geometric base is defined as a finite mm-space:

$$B_k = ([M], d_k, \mu_k), \tag{2}$$

where the metric component $d_k$ is represented by a learnable matrix $\mathbf{B}_k \in [0,1]^{M \times M}$. $\mathbf{B}_k$ is symmetric, hollow, and bounded, and serves as the concrete parameterization of the geometric base. Following common practice in GW-based learning, we do not enforce the triangle inequality, as pseudo-metrics remain valid distance kernels. The measure $\mu_k$ is fixed to a uniform distribution over the $M$ points, which avoids introducing unnecessary degrees of freedom and ensures well-defined GW couplings.

**Graph as a Metric Measure Space.** We denote each input graph as $G = (\mathcal{V}, \mathcal{E}, \mathbf{A}, \mathbf{X})$, with node set $\mathcal{V}$, edge set $\mathcal{E}$, adjacency matrix $\mathbf{A}$, and node features $\mathbf{X} \in \mathbb{R}^{N \times F}$ ($N = |\mathcal{V}|$, $F$ is the feature dimension). We then convert $G$ into a mm-space:

$$\mathcal{G} = (\mathcal{V}, d_G, \mu_G), \tag{3}$$

where $d_G$ creates the structure of graph, represented here by $\mathbf{A}$. $\mu_G$ is a **degree-based measure**:

$$\mu_G(i) = \frac{\deg(i)}{\sum_{v \in V} \deg(v)}. \tag{4}$$

where $\deg(i)$ is the degree of i-th node.

**GW-based Structural Alignment.** For each graph $G_i$, we compute its GW discrepancy with each base:

$$\delta_k = d_{GW}(\mathbf{A}_i, \mathbf{B}_k), \tag{5}$$

approximated using **Sliced Gromov-Wasserstein (SGW)** (Vayer et al., 2019). Crucially, by projecting the metric structure onto 1D slices, SGW reduces the computational bottleneck from the prohibitive cubic complexity $\mathcal{O}(N^3)$ of exact solvers to a **quasi-linear** $\mathcal{O}(N \log N)$ cost. This efficiency is fundamental to our method's scalability.

**Linear Surrogate of GW Barycenter.** Computing the true GW barycenter

$$\arg \min_B \mathbb{E}_{\mathcal{G}} \left[ d_{GW}(\mathcal{G}, B) \right], \tag{6}$$

is intractable because barycenter computation requires nested Optimal Transport (OT) optimization (Peyré et al.,

2016). We thus adopt a **linear surrogate model**:

$$\widetilde{\mathbf{B}}(G) = \sum_{k=1}^{K} w_k \mathbf{B}_k, \tag{7}$$

where $K$ denotes the number of geometric bases. The weight vector $\mathbf{w}$, which serves as the **structural coordinates** of $G$, is computed based on structural similarity:

$$w_k = \frac{\exp(-\delta_k/\tau)}{\sum_j \exp(-\delta_j/\tau)}. \tag{8}$$

where $\tau$ is a temperature hyperparameter.

**Theorem 3.2 (Stability of Structural Coordinates).** *Let $G$ and $G'$ be two graphs with bounded GW distance $d_{GW}(\mathcal{G}, \mathcal{G}') \leq \eta, 0 \leq \eta < \infty$. The structural coordinates $\mathbf{w}$ and $\mathbf{w}'$ computed via Eq. 8 is Lipschitz continuous with respect to the GW distance:*

$$\|\mathbf{w} - \mathbf{w}'\|_2 \leq L_w \cdot \eta, \tag{9}$$

*where $L_w = \frac{\sqrt{K} L_{sm}}{\tau}$ is the Lipschitz constant and $L_{sm}$ is the Lipschitz constant of the softmax function over a bounded domain. We provide the detailed proof in Appendix H.1.*

**Reconstruction of Graph-level Structure.** Linear combination is not a literal mm-space barycenter, but an efficient approximation of a GW barycenter expansion through a finite basis dictionary. Consequently, the structural reconstruction loss is formulated as:

$$\mathcal{L}_{\text{gw}} = \mathbb{E}_G \left[ d_{GW}(\mathbf{A}, \widetilde{\mathbf{B}}(G)) \right]. \tag{10}$$

**Reconstruction of Graph-level Statistics.** To ensure that structural coordinates preserve key graph-level statistics, we decode $\mathbf{w}$ using a Multi-Layer Perceptron (MLP) decoder $f(\cdot) \in \mathbb{R}^r$. The reconstruction target includes degree histogram, clustering histogram, and log-scaled motifs (triangles and short cycles). The reconstruction loss is:

$$\mathcal{L}_{\text{rec}} = \text{MSE}(\text{FE}(G), f(\mathbf{w})), \tag{11}$$

where $\text{FE}(\cdot)$ denotes the feature-extraction operator that computes graph-level statistics.

**Corollary 3.3 (Stability of Statistics Reconstruction).** *Following Theorem 3.2, the reconstructed graph-level statistics $f(\mathbf{w})$ is Lipschitz continuous with respect to the GW distance:*

$$\|f(\mathbf{w}) - f(\mathbf{w}')\|_2 \leq L_f \|\mathbf{w} - \mathbf{w}'\|_2 \leq L_f L_w \cdot \eta. \tag{12}$$

*where $L_f$ is the Lipschitz constant of the decoder $f(\cdot)$. This guarantees that structurally similar graphs produce similar reconstructed statistical descriptors, thereby preserving intrinsic geometric semantics.*

**Regularization of Structural Diversity.** To prevent bases from collapsing, we enforce diversity by minimizing the pairwise similarity of the learnable geometric bases $\mathbf{B}_k$:

$$\mathcal{L}_{\text{div}} = \frac{1}{|\mathcal{P}|} \sum_{(i,j)\in\mathcal{P}} \max(0, m - \|\mathbf{B}_i - \mathbf{B}_j\|_{\text{F}}), \quad (13)$$

where $\mathcal{P} = \{(i,j)|1 \leq i < j \leq K\}$ represents the set of all unique pairs of bases and $\|\cdot\|_{\text{F}}$ denotes the Frobenius norm. This loss enforces a minimum separation of $m$ between any pair of bases, ensuring that the geometric bases span a broad region of the structural space.

**Total Objective.** The final training objective is a weighted sum of the reconstruction loss and the regularization terms:

$$\mathcal{L}_{\text{total}} = \mathcal{L}_{\text{gw}} + \alpha\mathcal{L}_{\text{rec}} + \beta\mathcal{L}_{\text{div}}, \quad (14)$$

where $\alpha$ and $\beta$ are trade-off hyperparameters.

### 3.3. Downstream Projection

To derive a unified graph representation using pre-trained geometric bases, we project each input graph into a shared geometric coordinate system and aggregate the projected structural and feature information.

**Coordinate Projection.** For a given input graph $G_i$, we directly compute its structural coordinates $\mathbf{w}$ with respect to the pre-trained geometric bases $B$, strictly following the definition in Eq. 8. This maps the input graph to the shared geometric coordinate system.

**Feature Projection.** Using the OT plan $\mathbf{T}_{ik} \in \mathbb{R}^{N \times M}$ obtained during the GW distance computation between $(G_i, B_k)$ to project node features onto the geometric bases. To adhere to the injectivity requirements for multiset representations (Xu et al., 2019), we adopt summation aggregation. Specifically, since $\mathbf{T}_{ik}$ operates as a normalized measure (inducing averaging), we rescale the projection by $N$ to restore signal magnitude:

$$\mathbf{H}_k = N * \mathbf{T}_{ik}^{\top}\mathbf{X}_i \in \mathbb{R}^{M \times F},$$
$$\mathbf{H}(G_i) = \sum_{k=1}^{K} w_k\mathbf{H}_k \in \mathbb{R}^{M \times F}. \quad (15)$$

Finally, we construct the final graph embedding $\mathbf{z}(G_i)$ by concatenating the structural coordinates $\mathbf{w}$, reconstruction of graph-level statistics and the projected features:

$$\mathbf{z}(G_i) = [\mathbf{w} \,\|\, f(\mathbf{w}) \,\|\, \text{vec}(\mathbf{H}(G_i))] \in \mathbb{R}^{K+r+MF}. \quad (16)$$

Note that we use the frozen pre-trained decoder $f(\cdot)$.

## 4. Experiments

In this section, we evaluate the performance of SCGFM on few-shot classification tasks. Specifically, we aim to answer the following research questions: (1) **RQ1 (Performance):** Does SCGFM achieve superior generalization capabilities in few-shot learning, encompassing both in-domain and cross-domain transfers? (2) **RQ2 (Ablation and Scalability):** How do individual components and key design choices contribute to the effectiveness of SCGFM, and can the proposed design remain scalable to large graphs? (3) **RQ3 (Interpretability):** What do the learned geometric bases look like, and do they possess meaningful physical or topological interpretations? (4) **RQ4 (Mapping and Retrieval Quality):** How well does the learned latent space preserve the intrinsic geometric and neighborhood structure of the input graphs?

### 4.1. Experimental Setup

**Datasets and Tasks.** We evaluate cross-domain generalization on **12 datasets** from **5 domains**, covering both *graph classification* and *node classification*. Graph classification includes COX2, NCI1, BZR (**Molecules**), COLLAB, IMDB-BINARY (**Social Networks**), and COLORS-3 (**Synthetic**). Node classification includes Cora, CiteSeer, PubMed (**Academic**), Photo, Computers (**E-commerce**), and Reddit (**Social Networks**). Dataset statistics are provided in Appendix D.

**Evaluation Protocol.** We adopt a unified few-shot transfer learning paradigm: models are pre-trained on the source domain and directly evaluated on target domains with a **fixed encoder**. Downstream performance is measured using **Prototypical Network (PN)** for classification.

**Node Classification.** We formulate node classification as a graph classification task via node-centric subgraph sampling. Adopting the **Personalized Page Rank (PPR)** sampling strategy (Zeng et al., 2020), we extract subgraphs (size $\leq$ 100) centered on target nodes, which inherit the central node's label. Details are deferred to Appendix E.

**Baselines.** We compare SCGFM with: (1) **Vanilla GNNs:** GCN (Kipf & Welling, 2017), GIN (Xu et al., 2019), GAT (Veličković et al., 2018); (2) **Self-supervised methods:** DGI (Velickovic et al., 2019), GraphCL (You et al., 2020), GraphMAE (Hou et al., 2022), GraphACL (Xiao et al., 2023), S2GAE (Tan et al., 2023); (3) **GFMs:** GCOPE (Sun et al., 2024), RiemannGFM (Sun et al., 2025), and GIT (Wang et al., 2025g). For fair comparison, all pre-training methods adopt the same two-layer GCN backbone unless otherwise specified.

### 4.2. RQ1: Few-Shot Node and Graph Classification

We evaluate SCGFM under a unified 5-shot, fixed-encoder protocol on both graph-level (Table 1) and node-level (Table 2) transfer benchmarks. Overall, SCGFM consistently outperforms strong Self-supervised Learning (SSL) base-

*Table 1.* Accuracy (% ± standard deviation for 50 runs ) of 5-shot graph classification. OOD is Out-of-Domain. COL-3 = COLORS-3. IMDB-B = IMDB-BINARY. S1 = COX2 + NCI1 + BZR. S2 = COLLAB + IMDB-B. The best results are shown in **bold** and the runner-ups are underlined.

| Method | In-Domain | | | Trained on S1 (OOD) | | | Trained on S2 (OOD) | | | | Avg. |
|---|---|---|---|---|---|---|---|---|---|---|---|
| | COX2 | NCI1 | BZR | COLLAB | IMDB-B | COL-3 | COL-3 | COX2 | NCI1 | BZR | |
| GCN (Kipf & Welling, 2017) | $49.84_{\pm4.24}$ | $51.85_{\pm4.46}$ | $54.41_{\pm6.93}$ | $46.35_{\pm6.72}$ | $51.21_{\pm8.59}$ | $9.53_{\pm1.06}$ | $9.37_{\pm0.95}$ | $50.33_{\pm3.74}$ | $51.75_{\pm5.52}$ | $57.66_{\pm5.45}$ | 43.23 |
| GAT (Veličković et al., 2018) | $52.05_{\pm4.29}$ | $52.36_{\pm5.02}$ | $55.26_{\pm5.39}$ | $32.82_{\pm3.12}$ | $49.60_{\pm2.92}$ | $9.12_{\pm0.83}$ | $9.38_{\pm0.98}$ | $51.42_{\pm5.01}$ | $51.98_{\pm5.79}$ | $55.29_{\pm6.88}$ | 41.93 |
| GIN (Xu et al., 2019) | $54.31_{\pm6.36}$ | $52.95_{\pm4.77}$ | $51.29_{\pm4.67}$ | $58.11_{\pm4.48}$ | $55.36_{\pm4.28}$ | $9.25_{\pm1.10}$ | $9.12_{\pm1.03}$ | $\mathbf{55.16}_{\pm7.05}$ | $51.60_{\pm5.64}$ | $51.44_{\pm5.28}$ | 44.85 |
| DGI (Velickovic et al., 2019) | $49.34_{\pm5.34}$ | $50.86_{\pm4.85}$ | $50.30_{\pm6.10}$ | $51.77_{\pm7.06}$ | $54.26_{\pm9.18}$ | $9.67_{\pm1.17}$ | $9.77_{\pm1.21}$ | $49.40_{\pm5.18}$ | $50.60_{\pm4.69}$ | $49.98_{\pm6.94}$ | 42.60 |
| GraphCL (You et al., 2020) | $54.68_{\pm7.47}$ | $57.22_{\pm9.75}$ | $60.28_{\pm8.73}$ | $41.95_{\pm7.48}$ | $48.26_{\pm4.78}$ | $9.17_{\pm1.43}$ | $9.25_{\pm2.19}$ | $50.14_{\pm6.06}$ | $56.78_{\pm10.23}$ | $55.50_{\pm6.28}$ | 43.90 |
| GraphMAE (Hou et al., 2022) | $55.16_{\pm11.46}$ | $52.27_{\pm4.06}$ | $56.58_{\pm12.62}$ | $36.76_{\pm7.53}$ | $56.35_{\pm6.48}$ | $13.74_{\pm1.02}$ | $9.07_{\pm0.49}$ | $55.09_{\pm9.33}$ | $50.30_{\pm2.89}$ | $54.52_{\pm10.65}$ | 43.98 |
| GraphACL(Xiao et al., 2023) | $\underline{51.70}_{\pm6.91}$ | $50.90_{\pm5.16}$ | $57.72_{\pm8.70}$ | $\mathbf{63.59}_{\pm3.48}$ | $\underline{53.79}_{\pm6.55}$ | $11.91_{\pm1.37}$ | $9.13_{\pm0.89}$ | $\underline{51.00}_{\pm6.68}$ | $50.44_{\pm5.43}$ | $51.50_{\pm4.85}$ | 45.17 |
| S2GAE (Tan et al., 2023) | $53.19_{\pm5.11}$ | $52.14_{\pm4.95}$ | $55.83_{\pm6.96}$ | $54.79_{\pm7.14}$ | $50.48_{\pm4.03}$ | $\underline{14.47}_{\pm1.66}$ | $15.65_{\pm2.02}$ | $52.51_{\pm4.53}$ | $51.75_{\pm4.92}$ | $56.10_{\pm6.84}$ | $\underline{45.70}$ |
| GCOPE (Sun et al., 2024) | $53.20_{\pm9.06}$ | $51.75_{\pm10.31}$ | $56.90_{\pm8.92}$ | $61.07_{\pm7.61}$ | $51.60_{\pm9.07}$ | $9.16_{\pm1.81}$ | $8.95_{\pm1.58}$ | $50.60_{\pm6.55}$ | $51.95_{\pm7.47}$ | $58.10_{\pm10.22}$ | 45.32 |
| RiemannGFM (Sun et al., 2025) | $50.78_{\pm1.41}$ | $50.84_{\pm2.18}$ | $60.60_{\pm1.75}$ | $61.52_{\pm2.64}$ | $56.32_{\pm3.06}$ | $8.88_{\pm1.00}$ | $10.36_{\pm0.87}$ | $50.36_{\pm1.77}$ | $51.28_{\pm2.19}$ | $58.24_{\pm3.04}$ | 44.92 |
| GIT (Wang et al., 2025g) | $51.54_{\pm5.31}$ | $50.70_{\pm6.52}$ | $51.86_{\pm8.65}$ | $49.32_{\pm12.17}$ | $54.62_{\pm8.44}$ | $9.37_{\pm1.09}$ | $9.04_{\pm1.45}$ | $49.30_{\pm4.08}$ | $52.12_{\pm6.72}$ | $\underline{52.46}_{\pm6.88}$ | 43.03 |
| SCGFM (Ours) | $\mathbf{56.60}_{\pm6.12}$ | $\mathbf{58.00}_{\pm8.90}$ | $\mathbf{60.80}_{\pm6.60}$ | $\underline{61.93}_{\pm6.11}$ | $\mathbf{56.62}_{\pm9.00}$ | $\mathbf{26.31}_{\pm2.99}$ | $\mathbf{26.97}_{\pm3.08}$ | $53.80_{\pm5.61}$ | $\mathbf{57.50}_{\pm7.32}$ | $\mathbf{58.50}_{\pm6.28}$ | **51.70** |

*Table 2.* Accuracy (% ± standard deviation for 50 runs ) of 5-shot node classification. N1 = Cora + CiteSeer + PubMed. N2 = Photo + Computers. The best results are shown in **bold** and the runner-ups are underlined.

| Method | In-Domain | | | Trained on N1 (OOD) | | | Trained on N2 (OOD) | | | | Avg. |
|---|---|---|---|---|---|---|---|---|---|---|---|
| | Cora | CiteSeer | PubMed | Photo | Computers | Reddit | Cora | CiteSeer | PubMed | Reddit | |
| GCN (Kipf & Welling, 2017) | $36.01_{\pm3.33}$ | $27.21_{\pm3.88}$ | $44.08_{\pm5.32}$ | $41.21_{\pm4.20}$ | $28.00_{\pm3.35}$ | $18.79_{\pm1.26}$ | $47.13_{\pm2.26}$ | $30.19_{\pm4.18}$ | $42.57_{\pm5.76}$ | $44.54_{\pm1.36}$ | 35.97 |
| GAT (Veličković et al., 2018) | $39.47_{\pm3.89}$ | $30.65_{\pm4.21}$ | $45.68_{\pm5.58}$ | $35.60_{\pm3.39}$ | $31.01_{\pm3.19}$ | $13.87_{\pm1.65}$ | $48.38_{\pm3.18}$ | $35.38_{\pm4.31}$ | $52.56_{\pm6.89}$ | $20.60_{\pm1.59}$ | 35.32 |
| GIN (Xu et al., 2019) | $39.23_{\pm3.70}$ | $24.60_{\pm4.71}$ | $41.48_{\pm7.10}$ | $39.48_{\pm4.84}$ | $32.52_{\pm2.89}$ | $19.98_{\pm1.79}$ | $32.55_{\pm3.01}$ | $25.68_{\pm3.84}$ | $34.77_{\pm5.65}$ | $22.26_{\pm1.49}$ | 31.26 |
| DGI (Velickovic et al., 2019) | $54.61_{\pm6.75}$ | $28.50_{\pm6.42}$ | $54.60_{\pm7.86}$ | $58.44_{\pm5.56}$ | $42.18_{\pm4.54}$ | $42.14_{\pm2.38}$ | $50.67_{\pm6.00}$ | $25.13_{\pm5.28}$ | $51.07_{\pm8.33}$ | $62.60_{\pm2.55}$ | 47.00 |
| GraphCL (You et al., 2020) | $60.82_{\pm6.38}$ | $35.50_{\pm5.81}$ | $57.19_{\pm8.72}$ | $50.95_{\pm3.12}$ | $30.44_{\pm3.93}$ | $67.75_{\pm1.58}$ | $59.31_{\pm3.02}$ | $38.53_{\pm4.47}$ | $43.17_{\pm5.57}$ | $67.56_{\pm1.15}$ | $\underline{51.12}$ |
| GraphMAE (Hou et al., 2022) | $38.50_{\pm5.38}$ | $\underline{34.32}_{\pm4.94}$ | $36.83_{\pm4.36}$ | $31.57_{\pm6.45}$ | $31.86_{\pm6.71}$ | $13.39_{\pm1.12}$ | $\underline{34.59}_{\pm4.33}$ | $\underline{28.45}_{\pm4.51}$ | $34.55_{\pm3.71}$ | $\underline{32.21}_{\pm1.62}$ | 31.63 |
| GraphACL(Xiao et al., 2023) | $48.00_{\pm5.44}$ | $34.02_{\pm5.86}$ | $35.70_{\pm7.54}$ | $59.51_{\pm4.94}$ | $51.48_{\pm4.85}$ | $44.91_{\pm2.22}$ | $34.79_{\pm4.57}$ | $26.00_{\pm5.69}$ | $35.17_{\pm7.02}$ | $45.94_{\pm2.23}$ | 41.56 |
| S2GAE (Tan et al., 2023) | $66.83_{\pm5.53}$ | $32.55_{\pm6.38}$ | $58.90_{\pm8.35}$ | $61.13_{\pm4.66}$ | $65.59_{\pm3.96}$ | $75.28_{\pm1.65}$ | $33.73_{\pm5.21}$ | $23.73_{\pm5.21}$ | $39.77_{\pm7.91}$ | $40.95_{\pm1.94}$ | 49.87 |
| GCOPE (Sun et al., 2024) | $47.84_{\pm4.79}$ | $30.13_{\pm6.00}$ | $43.23_{\pm6.54}$ | $60.92_{\pm4.54}$ | $53.70_{\pm3.90}$ | $55.72_{\pm2.02}$ | $43.43_{\pm6.05}$ | $26.08_{\pm4.82}$ | $42.23_{\pm7.74}$ | $23.00_{\pm2.36}$ | 42.63 |
| RiemannGFM (Sun et al., 2025) | $51.09_{\pm1.10}$ | $31.95_{\pm1.44}$ | $40.96_{\pm2.25}$ | $48.97_{\pm1.10}$ | $31.44_{\pm1.05}$ | $33.51_{\pm0.41}$ | $46.27_{\pm1.23}$ | $31.61_{\pm1.17}$ | $39.07_{\pm1.59}$ | $18.65_{\pm0.28}$ | 37.35 |
| GIT (Wang et al., 2025g) | $32.01_{\pm5.39}$ | $28.75_{\pm4.71}$ | $36.83_{\pm4.36}$ | $42.86_{\pm6.20}$ | $35.76_{\pm7.40}$ | $33.50_{\pm5.76}$ | $34.91_{\pm4.68}$ | $26.35_{\pm4.36}$ | $34.65_{\pm3.21}$ | $32.90_{\pm5.57}$ | 33.85 |
| SCGFM (Ours) | $\mathbf{70.54}_{\pm3.27}$ | $\mathbf{43.83}_{\pm4.26}$ | $\mathbf{68.19}_{\pm5.30}$ | $\mathbf{68.30}_{\pm3.93}$ | $\underline{57.35}_{\pm4.22}$ | $\mathbf{84.17}_{\pm1.23}$ | $\mathbf{70.55}_{\pm3.28}$ | $\mathbf{43.82}_{\pm4.26}$ | $\mathbf{68.15}_{\pm5.26}$ | $\mathbf{84.16}_{\pm1.23}$ | **65.90** |

lines and recent GFM-style methods, showing superior robustness across domains and tasks.

**Graph Classification.** SCGFM achieves strong in-domain performance on COX2, NCI1, and BZR, and maintains clear advantages in cross-domain transfer. In particular, on COLORS-3—characterized by large structural shifts and an 11-way label space—most baselines degrade to near chance-level accuracy, while SCGFM remains substantially more robust. This gain is not solely driven by COLORS-3: excluding its two transfer columns, SCGFM still attains a mean accuracy of $57.82\%$, outperforming the suboptimal RiemannGFM by $4.08\%$.

**Source-independent Stability in Node Classification.** A striking observation from Table 2 is that SCGFM exhibits near source-independent behavior in node classification. For example, on Reddit, performance remains consistently high ($\approx 84.17\%$) regardless of the encoder's pre-training domain.

To better understand this phenomenon, we analyze the similarity of target-domain embeddings produced by encoders pre-trained on different source domains (Figure 3). We report **Centered Kernel Alignment (CKA** $\in [0, 1]$) (Kornblith et al., 2019) and the **Pearson correlation coefficient** ($\rho \in [-1, 1]$) to quantify representation similarity across source domains, where larger values in both metrics indi-

cate stronger agreement between the resulting embedding spaces.

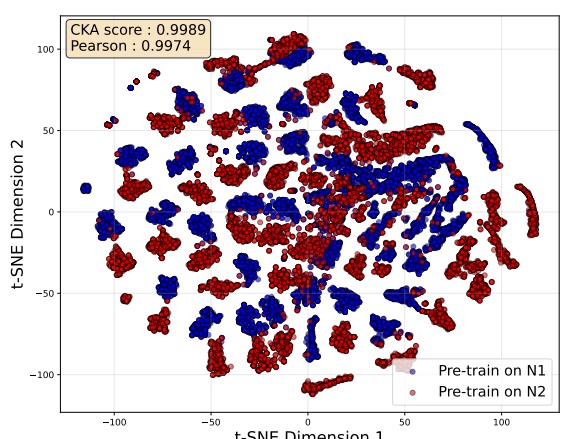

*Figure 3.* **Representation similarity across source domains on Reddit.** The t-SNE (Maaten & Hinton, 2008) visualization of node embeddings generated by encoders pre-trained on two different source domains.

Despite being trained on distinct sources, the resulting representations on Reddit display highly consistent latent geometries. This suggests that SCGFM learns a dominant semantic structure in the embedding space that is largely

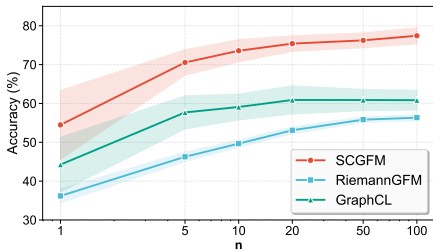

**Figure 4.** **n-shot node classification on Cora.** The colored region denotes the variance of the results.

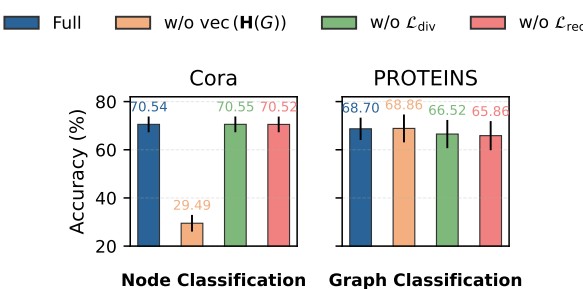

**Figure 5.** Ablation studies on Cora and PROTEINS (5-shot).

invariant to the choice of pre-training domain.

This contrasts with the graph classification results in Table 1, where domain shifts lead to noticeable performance variations. We attribute this difference to the dominance of high-dimensional semantics. Unlike graph classification datasets (e.g., molecular graphs), which rely heavily on structural topology, node classification benchmarks are rich in semantic features. SCGFM projects these features into an ultra–high-dimensional space ($\approx$45k dimensions on Cora with $M = 32$). According to **Cover's Theorem** (Cover, 1965), such a projection greatly increases the likelihood of linear separability, causing the decision boundary to be driven primarily by the intrinsic geometry of the semantic features rather than the pre-training domain.

Consequently, node classification performance is largely invariant to the pre-training source, and the remaining variance is dominated by task sampling effects rather than instability in the learned representations.

**Impact of Shot Number.** To assess robustness, we extend the evaluation across varying support sizes $n \in \{1, 5, \ldots, 100\}$. As shown in Figure 4, SCGFM consistently outperforms both baselines compared, with the largest margin observed in the extreme 1-shot regime, validating the effectiveness of our geometric inductive bias under severe data scarcity. Moreover, performance improves steadily with increasing supervision, indicating that SCGFM refines class representations without overfitting to outlier-induced variance.

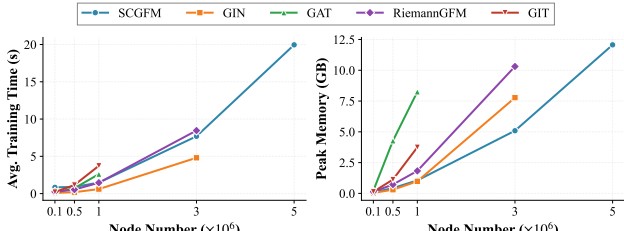

**Figure 6.** Scalability comparison on synthetic datasets.

### 4.3. RQ2: Ablation and Scalability

**Ablation Analysis.** As illustrated in Figure 5, the contribution of individual components varies markedly across data modalities, revealing a mechanism of adaptive reliance. On **Cora**, which features high-dimensional and informative node attributes, the learned representation is dominated by structure-aligned semantics $\text{vec}(\mathbf{H}(G))$. As expected, removing this component leads to a catastrophic performance drop ($70.54\% \rightarrow 29.49\%$), indicating that geometric coordinates alone are insufficient for content-heavy tasks.

In contrast, on the structure-centric **PROTEINS** dataset, removing $\text{vec}(\mathbf{H}(G))$ yields a marginal performance improvement ($68.70\% \rightarrow 68.86\%$).We attribute this counterintuitive effect to semantic interference: when labels are primarily determined by intrinsic topology rather than node attributes, injecting weak or misaligned semantic signals ($\mathbf{H}(G)$) introduces noise that dilutes the discriminative power of the geometric coordinates $\mathbf{w}$.

Despite this modality-dependent dichotomy, the optimization constraints are universally critical. Removing either the reconstruction loss $\mathcal{L}\text{rec}$ or the diversity regularizer $\mathcal{L}\text{div}$ consistently degrades performance across all datasets, confirming their necessity for learning a stable and non-degenerate geometric basis space.

**Scalability.** We evaluate scalability on synthetic datasets consisting of 1k graphs, with the average graph size varying up to 5k nodes. As shown in Figure 6, SCGFM scales smoothly to $5 \times 10^6$ total nodes while maintaining the lowest peak memory consumption ($\approx$12 GB) among all methods. In contrast, **GAT** and **GIT** encounter out-of-memory (OOM) errors at $3 \times 10^6$ nodes due to their high computational overhead. At $5 \times 10^6$ nodes, both **GIN** and **RiemannGFM** exceed memory limits because of their substantially higher memory usage.

**Pre-training Computational Analysis.** The computational bottleneck of SCGFM lies in the SGW-based metric alignment during pre-training. By replacing exact graph matching with sliced projections, the complexity scales as $O(KL(N \log N + M \log M))$, which reduces to $O(N \log N)$ in practice since the number of bases $K$, the number of slices $L$, and base size $M$ are small constants. All

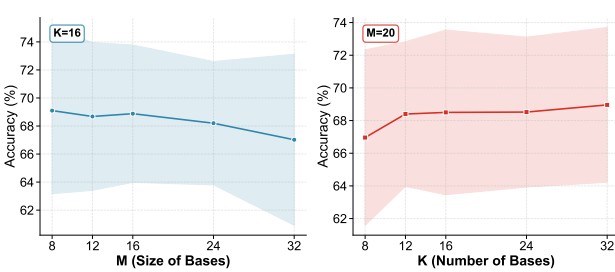

*Figure 7.* Hyperparameter analysis on PROTEINS (5-shot).

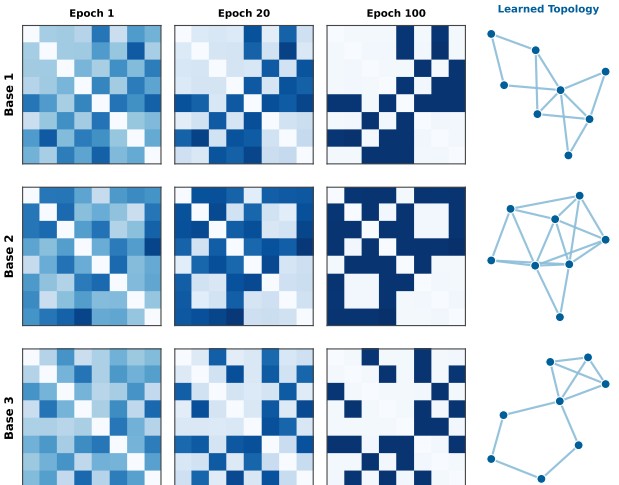

*Figure 8.* **Evolution of learned geometric bases on MUTAG.** Rows represent three distinct learned bases; columns display the pseudo-metric heatmaps during training (Epoch 1, 20, 100) and the final induced topology.

remaining components introduce negligible overhead, while feature statistics $FE(G)$ are computed once as an offline preprocessing step.

**Hyperparameters Sensitivity Analysis.** We analyze the impact of the hyperparameters $K$ and $M$ on the performance of SCGFM (Figure 7). Increasing the number of bases $K$ improves accuracy up to $K = 12$, beyond which performance plateaus around $68.5\% \rightarrow 69.0\%$. This suggests that a compact dictionary of 12–16 bases is sufficient to capture the underlying graph manifold in PROTEINS. SCGFM is also robust to the choice of base size $M$, exhibiting stable performance for $M \in [8, 16]$, which aligns with the typical scale of functional motifs in protein graphs. Larger bases ($M = 32$) introduce unnecessary complexity and slightly degrade performance.

### 4.4. RQ3: Interpretability of Geometric Bases

To evaluate the interpretability of the learned geometric bases, we visualize the training dynamics of three bases on the MUTAG dataset (Figure 8). The pseudo-metric

*Table 3.* Domain-dependent activation of learned geometric bases. The uniform activation weight is $1/8 = 0.125$.

| DOMAIN | $B_7$ BIO. | $B_6$ SOC. | $B_2/B_8$ SHARED |
|---|---|---|---|
| PROTEINS | **0.1803** | 0.1265 | $\sim 0.178$ |
| IMDB-B | 0.1023 | **0.1719** | $\sim 0.166$ |
| SHIFT | $1.76\times$ | $1.36\times$ | $1.07$ |

*Table 4.* Topological complexity of learned bases under different base sizes $M$.

| $M$ | Avg. Degree | Avg. Comp. | Topology Focus |
|---|---|---|---|
| 8 | 2.61 | 1.62 | Simple primitives |
| 16 | 4.71 | 1.06 | Connected subgraphs |
| 24 | 6.60 | 1.04 | Complex motifs |

heatmaps evolve from diffuse, unstructured patterns at initialization to sharp, high-contrast block structures after convergence, indicating that the model progressively suppresses noise and captures consistent structural relations.

The reconstructed graphs reveal that different bases specialize in distinct geometric motifs. Specifically, **Base 2** encodes a dense, highly clustered topology resembling aromatic-like substructures, whereas **Base 3** disentangles a cyclic core with a branching path, forming a recognizable ring-and-tail pattern. Overall, the learned bases display clear specialization and diversity, demonstrating that SCGFM discovers meaningful geometric primitives rather than collapsing to a single trivial topology.

**Base specialization across diverse domains.** We jointly pretrain SCGFM on PROTEINS from bioinformatics and IMDB-BINARY from social networks using $K = 8$ bases, without providing any domain labels. We then compute the average activation weight $w_k$ of each base on graphs from the two domains. The results are summarized in Table 3.

The domain-specific shifts are statistically significant $p < 10^{-6}$. Bases $B_7$ and $B_6$ show clear domain preferences, corresponding respectively to bioinformatics-related and social-network-related structural motifs. In contrast, $B_2$ and $B_8$ maintain consistently high activation in both domains, suggesting that they capture universal geometric primitives shared across domains. These results indicate that SCGFM can naturally disentangle domain-specific motifs from domain-agnostic topological patterns without supervision.

**Scaling the base vocabulary.** We further analyze how learned bases change with the base size $M$ and the number of bases $K$. Table 4 shows that larger $M$ yields higher-resolution bases, moving from simple primitives to more connected and complex motifs. As $K$ increases (Table 5), the top-1 usage ratio drops from 75% to 33%, indicating

*Table 5.* Role differentiation of bases as the number of bases $K$ increases.

| $K$ | Unique Top-1 / $K$ | Usage Ratio |
|---|---|---|
| 4 | 3 / 4 | 75% |
| 8 | 4 / 8 | 50% |
| 16 | 9 / 16 | 56% |
| 24 | 8 / 24 | 33% |

that the learned vocabulary naturally splits into a few dominant prototypes and auxiliary bases used for soft structural refinement.

### 4.5. RQ4: Geometry Alignment, Retrieval, and Approximate Isometry

We investigate whether the learned geometric bases induce an approximately isometric embedding, effectively mapping the complex mm-space of graphs into a structurally meaningful Euclidean vector space. Specifically, our aim is to verify whether the Euclidean distances in the latent space consistently align with the intrinsic GW distances between graphs. As shown in Figure 9, the latent distances learned by SCGFM exhibit a strong and statistically significant linear correlation with the true GW distances on NCI1 ($\rho = 0.7005$, $P < 10^{-4}$). This quantitative result indicates that the latent space faithfully preserves the intrinsic geometry of graphs; that is, graph pairs with high topological similarity (low GW distance) are mapped to proximal points in the embedding space, while dissimilar graphs represent distant points. This finding confirms that SCGFM produces an approximately isometric embedding, where Euclidean distances accurately reflect intrinsic graph similarities.

**Graph Retrieval.** Beyond pairwise geometry alignment, we further evaluate whether the learned latent space supports retrieval-oriented downstream usage. Given a query graph, the database graphs are ranked according to their embedding distances, and we report Precision@10, MAP@10, and $k$NN classification accuracy at $k = 10$. As shown in Table 6, SCGFM consistently outperforms GCN, GAT, and GIN on both MUTAG and COX2. These results indicate that SCGFM embeddings preserve meaningful neighborhood structure, making graphs from the same semantic class closer in the learned geometric space.

## 5. Conclusion

We introduce SCGFM, a structure-centric graph foundation model designed to overcome structural heterogeneity and feature mismatching in cross-domain graph learning. By learning a set of geometric structural bases and leveraging Gromov-Wasserstein distances for projection, SCGFM projects heterogeneous graphs into a shared geometric space

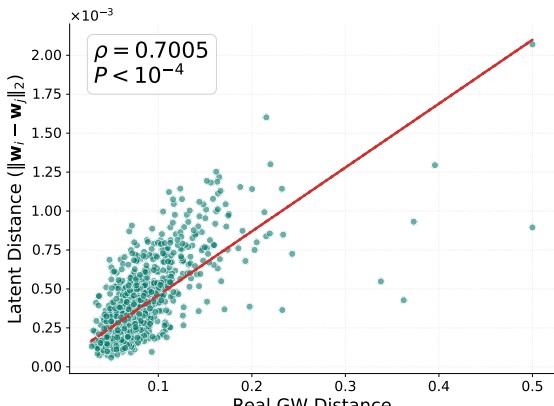

*Figure 9.* **Approximate isometry on NCI1.** Scatter plot of latent Euclidean distances versus true GW distances. The solid red line indicates the linear regression fit to the data. Pearson correlation ($\rho$) and P-value are reported in the figure.

*Table 6.* Graph retrieval performance using frozen embeddings.

| Dataset | Model | P@10 ↑ | MAP@10 ↑ | kNN@10 ↑ |
|---|---|---|---|---|
| MUTAG | SCGFM | **0.784** | **0.716** | **0.835** |
| | GIN | 0.736 | 0.666 | 0.798 |
| | GCN | 0.669 | 0.582 | 0.739 |
| | GAT | 0.654 | 0.551 | 0.734 |
| COX2 | SCGFM | **0.702** | **0.607** | **0.792** |
| | GCN | 0.675 | 0.570 | 0.788 |
| | GAT | 0.674 | 0.568 | 0.779 |
| | GIN | 0.667 | 0.552 | 0.773 |

that cleanly decouples and aligns topology with semantics. Extensive experiments show that SCGFM consistently surpasses state-of-the-art self-supervised methods and existing GFMs on few-shot node-and graph-level classification tasks across domains. Beyond empirical gains, SCGFM demonstrates strong cross-domain generalization, robustness, and scalability, while providing theoretical guarantees on the stability of the structural coordinates and the reconstructed representations.

## Impact Statement

This work advances graph foundation models by improving cross-domain structural representation learning. We do not identify any specific ethical concerns beyond those commonly associated with machine learning research.

## Acknowledgements

This work was supported by the National Natural Science Foundation of China (No. 62276053). This work was supported in part by Sichuan Province Science and Technology Support Program under Grant No.2025ZDZX0016.

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

# A. Notations

*Table 7.* Summary of Notations.

| Symbol | Description |
|--------|-------------|
| $G = (\mathcal{V}, \mathcal{E}, \mathbf{A}, \mathbf{X})$ | An undirected graph with node set $\mathcal{V}$, edge set $\mathcal{E}$. Adjacency matrix $\mathbf{A}$ and feature matrix $\mathbf{X}$. |
| $|\mathcal{V}|, N$ | Number of nodes in graph $G$. |
| $|\mathcal{E}|$ | Number of edges in graph $G$. |
| $\mathbf{A} \in \mathbb{R}^{N \times N}$ | Adjacency matrix of graph $G$. |
| $\mathbf{X} \in \mathbb{R}^{N \times F}$ | Node feature matrix. |
| $F$ | Dimension of node feature. |
| $\mathcal{G}$ | mm-space representation of input graph. $G$ |
| $d_G$ | Metric of the graph $\mathcal{G}$. |
| $\mu_G$ | Probability measure of the graph mm-space $\mathcal{G}$. |
| $\mathbf{q}$ | R Coordinate representation of G |
| $K$ | Number of geometric bases. |
| $\mathbf{B}_k \in \mathbb{R}^{M \times M}$ | k-th geometric base matrix. |
| $B_k$ | mm-space representation of the $k$-th geometric base. |
| $M$ | Number of nodes in geometric base. |
| $d_k$ | The metric component of the $k$-th geometric base $B_k$. |
| $\mu_k$ | The probability measure associated with the $k$-th geometric base $B_k$. |
| $\mathbf{w}$ | Weight vector base on structural similarity between input graph $G$ and geometric bases. |
| $\mathbf{T}_{ik} \in \mathbb{R}^{N \times M}$ | Optimal transport plans between graph $G_i$ and geometric base $\mathbf{B}_k$. |
| $L$ | Number of slices in SGW computation. |
| $\tau$ | Temperature parameter. |
| $d_{GW}(G, G')$ | GW distance between graph $G$ and $G'$. |
| $\eta$ | Upperbound of GW distance. |
| $\boldsymbol{\delta} \in \mathbb{R}^K$ | Consists of the GW distances from graph $G$ to the set of fixed geometric bases |
| $\mathcal{X}$ | Space of mm-spaces equipped with the GW distance. |
| $\mathcal{K}$ | Totally bounded subset of $\mathcal{X}$. |
| $\mathcal{L}_{gw}$ | Loss about GW distance between graph $G$ and linear surrogate graph. |
| $\mathcal{L}_{rec}$ | Loss of feature reconstruction. |
| $\mathcal{L}_{div}$ | Loss of structural diversity. |
| $m$ | Diversity margin. |
| $f(\cdot)$ | 2-layer MLP decoder. |
| $\theta$ | Decoder Parameters. |
| $r$ | Output Dimension of $f(\cdot)$. |
| $L_w$ | Lipschitz constant for structural coordinates $\mathbf{w}$. |
| $L_{sm}$ | Lipschitz constant of softmax function $L_f$ Lipschitz constant of decoder $f(\cdot)$. |
| $\text{FE}(\cdot)$ | Feature-extraction operator computing graph-level statistics. |
| $\mathcal{P}$ | Represents the set of all unique pairs of bases. |
| $\alpha$ | trade-off hyperparameter about $\mathcal{L}_{rec}$. |
| $\beta$ | trade-off hyperparameter about $\mathcal{L}_{div}$. |
| $\mathbf{H}_k$ | k-th hidden representation. |
| $\mathbf{H}(G_i)$ | Projected feature matrix of graph $G_i$. |
| $\mathbf{z}(G_i)$ | Final graph embedding. |

# B. Geometric Bases Pre-training Pipeline

The pre-training pipeline of SCGFM is illustrated in Algorithm 1.

# C. Complexity Analysis of SCGFM.

We provide a more detailed analysis of SCGFM below.

**Training Complexity.** The main compuatational bottleneck lies in metric alignment during pre-training. We adopt **Sliced Gromov-Wasserstein (SGW)**, which reduces the dominant cost to

$$O\big(KL(N \log N + M \log M)\big),$$

---

**Algorithm 1** Pre-training pipeline of SCGFM

---

1: **Input:** Graph dataset $\mathcal{D} = \{G_1, \ldots, G_N\}$, geometric bases number $K$ with $M$ nodes, slicing number $L$. Hyperparameters $\tau, \alpha, \beta$.
2: **Output:** Pre-trained geometric bases $\mathcal{B} = \{\mathbf{B}_k\}_{k=1}^K$ and decoder $f_\theta(\cdot)$.
3: Initialize bases matrix $\mathbf{B}_k \in \mathbb{R}^{M \times M}$ and decoder parameters $\theta$ randomly;
4: Pre-compute feature statistics $\mathrm{FE}(G_i)$ for all $G_i \in \mathcal{D}$;
5: **for** epoch $e = 1, 2, \ldots, E$ **do**
6:     **for** each mini-batch batch $\mathcal{D}_t \subset \mathcal{D}$ **do**
7:         # 1. Metric Alignment (via Sliced GW)
8:         Calculate SGW discrepancy for each $G_i \in \mathcal{D}_t$: $\delta_k \leftarrow \mathrm{SGW}(\mathbf{A}_i, \mathbf{B}_k)$ via Eq. (5);
9:         Compute structural weights: $w_k \leftarrow \mathrm{softmax}(-\delta_k/\tau)$ via Eq. (8);
10:         # 2. Surrogate Construction & Alignment Loss
11:         Construct linear surrogate bases: $\tilde{\mathbf{B}}(G) \leftarrow \sum_{k=1}^K w_k \mathbf{B}_k$ via Eq. (7);
12:         Calculate GW alignment loss: $\mathcal{L}_{\mathbf{gw}} \leftarrow \mathrm{SGW}(\mathbf{A}_i, \tilde{\mathcal{B}}(G))$ via Eq. (10);
13:         # 3. Feature Reconstruction
14:         Decode weights to statistics: $\hat{\mathbf{s}} \leftarrow f_\theta(\mathbf{w})$;
15:         Calculate reconstruction loss: $\mathcal{L}_{\mathrm{rec}} \leftarrow \|\mathrm{FE}(G) - \hat{\mathbf{s}}\|_2^2$ via Eq. (11);
16:         # 4. Optimization with Diversity
17:         Calculate diversity regularization: $\mathcal{L}_{\mathrm{div}} \leftarrow \mathrm{PairwiseDist}(\mathcal{B})$ via Eq. (13);
18:         Total objective: $\mathcal{L}_{\mathrm{total}} \leftarrow \mathcal{L}_{\mathrm{gw}} + \alpha \mathcal{L}_{\mathrm{rec}} + \beta \mathcal{L}_{\mathrm{div}}$;
19:         Update $\mathcal{B}$ and $\theta$ by minimizing $\mathcal{L}_{\mathrm{total}}$;
20:     **end for**
21: **end for**

---

where $N$ and $M$ are the numbers of nodes in the input graph and geometric base, respectively, $K$ is the number of bases, and $L$ is the number of slices. Since $K$, $L$, and $M$ are small parameters independent of graph size, the resulting scaling is **near-linear** in practice.

**Inference Complexity.** The inference stage consists of three part:

- **Structural coordinate w computation** via SGW: $O\big(KL(N \log N + M \log M)\big)$

- **Nonlinear projection** $f(\mathbf{w})$: a lightweight MLP with negligible cost $O(K^2)$.

- **Feature projection H** via entropic GW: $O(NM \cdot iter)$, where $iter$ is the number of maximum iterations in entropic GW.

Therefore, the overall inference complexity is

$$O\big(KL(N \log N + M \log M) + K^2 + NM \cdot iter\big).$$

**Memory Complexity.** SCGFM stores only a compact set of geometric bases (e.g., $16 \times 20 \times 20$), and memory usage is dominated by the input graph adjacency plus the intermediate sliced projections. As a result, it achieves an overall memory complexity of $O(N + |E|)$. As further empirical evidence, the scalability experiment in Figure 6 demonstrates that SCGFM remains stable even under an extreme stress test with **5M nodes**, reaching a peak memory usage of approximately **12 GB**. Notably, SCGFM is the only method among the compared baselines that successfully completes the 5M-node setting.

Overall, both the theoretical analysis and the stress tests confirm that SCGFM achieves favorable scalability, owing to the SGW approximation and its compact geometric base design.

### C.1. Scalability with Respect to Geometric Bases

We further discuss the computational bottlenecks of SCGFM when the number of geometric bases or the basis size increases. As reported in the main paper, SCGFM shows strong scalability in both the scalability analysis and the sensitivity studies. Here we clarify this issue from theoretical and empirical perspectives.

First, the dominant training cost scales as

$$O\big(KL(N \log N + M \log M)\big), \tag{17}$$

where $K$ denotes the number of geometric bases, $M$ denotes the basis size, $N$ denotes the input graph size, and $L$ denotes the number of sliced projections. This indicates that increasing either $K$ or $M$ introduces approximately linear overhead in practice.

Second, the geometric bases are designed to serve as compact structural abstractions rather than large graph templates. When $M \ll N$, each basis acts as a compressed geometric primitive or motif that summarizes representative topological patterns. Increasing $M$ beyond a moderate range weakens this compression effect and may introduce redundant high-dimensional templates.

Third, our empirical results show that performance saturates once $K$ and $M$ reach modest values. To further validate this observation, we conduct a stress study on COLLAB by increasing either the number of bases or the bases size. The results are reported in Table 8.

Table 8. Scalability study of SCGFM with different numbers of bases $K$ and basis sizes $M$ on COLLAB.

| Setting | Variant | Time/Epoch (s) | Memory (GB) | 5-shot Acc. (%) |
|---|---|---|---|---|
| Default batch size $= 128$ | $K = 16, M = 32$ | $5.12 \pm 0.66$ | 0.23 | $66.40 \pm 6.33$ |
| Scale $K$ (fix $M = 32$) | $K = 32$ | $5.77 \pm 0.38$ | 0.34 | $66.12 \pm 6.12$ |
| | $K = 64$ | $5.76 \pm 0.35$ | 0.61 | $63.83 \pm 6.17$ |
| Scale $M$ (fix $K = 16$) | $M = 64$ | $4.85 \pm 0.11$ | 0.25 | $65.68 \pm 5.07$ |
| | $M = 128$ | $4.80 \pm 0.12$ | 0.37 | $64.53 \pm 4.69$ |

The results show that increasing the bases size to approximately four times the default configuration only introduces modest memory overhead, while the accuracy improvement is negligible or slightly negative. Similarly, increasing the number of bases does not consistently improve performance. These observations are consistent with the saturation trend reported in the main paper, suggesting that a compact geometric dictionary is sufficient to capture the structural diversity needed for effective transfer.

## D. Datasets

Detailed datasets is illustrated in Table 9

Table 9. Statistics of Multi-Domain Graph Datasets. NONE indicates the absence of input node features.

| Dataset | Domain | Graphs | # Nodes (or AVG.) | # Edges or AVG. | # Feature Dimensions | # Classes |
|---|---|---|---|---|---|---|
| COX2 (Sutherland et al., 2003) | Molecules | 467 | 41.22 | 43.45 | 3 | 2 |
| NCI1 (Wale et al., 2008) | Molecules | 4110 | 29.87 | 32.30 | NONE | 2 |
| BZR (Sutherland et al., 2003) | Molecules | 405 | 35.75 | 38.36 | 3 | 2 |
| MUTAG (Debnath et al., 1991) | Molecules | 188 | 17.93 | 19.79 | NONE | 2 |
| COLLAB (Yanardag & Vishwanathan, 2015) | Social Networks | 5000 | 74.49 | 2457.78 | NONE | 3 |
| IMDB-BINARY (Yanardag & Vishwanathan, 2015) | Social Networks | 1000 | 19.77 | 96.53 | NONE | 2 |
| COLORS-3 (Knyazev et al., 2019) | Synthetic | 10500 | 61.31 | 91.03 | 4 | 11 |
| PROTEINS (Borgwardt et al., 2005) | Bioinformatics | 1113 | 39.06 | 72.82 | 1 | 2 |
| Cora (Yang et al., 2016) | Academic | 1 | 2708 | 10,556 | 1,433 | 7 |
| CiteSeer (Yang et al., 2016) | Academic | 1 | 3,327 | 9104 | 3,703 | 6 |
| PubMed (Yang et al., 2016) | Academic | 1 | 19,717 | 88,648 | 500 | 3 |
| Photo (Shchur et al., 2018) | E-Commerce | 1 | 7,650 | 238,162 | 745 | 8 |
| Computers (Shchur et al., 2018) | E-Commerce | 1 | 13,752 | 491,722 | 767 | 10 |
| Reddit (Hamilton et al., 2017) | Social Networks | 1 | 232,965 | 114,615,892 | 602 | 41 |

# E. Experimental Setting Details

## E.1. Few-shot Learning Detailed Settings.

In the few-shot pretraining stage, all self-supervised models with configurable backbones adopt a two-layer GCN as the encoder to ensure architectural consistency across different methods. During the downstream few-shot evaluation, the pretrained encoder is frozen, and no further fine-tuning is performed. Instead, a prototypical network is employed as the sole classifier. This design choice allows us to isolate the quality of the representations learned during pretraining, eliminating the influence of task-specific adaptation.

All downstream few-shot tasks follow a unified evaluation protocol with **5-shot support sets, 50 query samples per class, and results averaged over 50 independent runs**. Such a strict few-shot setting is intentionally adopted to emphasize the transferability of cross-domain knowledge learned during pretraining, rather than the model's capacity to adapt through gradient-based updates. Moreover, using an identical evaluation protocol across all methods ensures a fair and reproducible comparison under limited supervision.

For node classification tasks, we transform the problem into a graph-level few-shot setting by constructing **ego-centric subgraphs using Personalized PageRank (PPR) sampling**. Specifically, for each target node, we sample a subgraph containing up to 100 nodes, and assign the label of the central node as the label of the corresponding subgraph. Importantly, the PPR-based subgraph sampling is performed *once* to construct a fixed dataset, which is shared by all compared methods. By ensuring that all models are trained and evaluated on exactly the same sampled subgraphs, we eliminate potential performance variations caused by stochastic sampling, thereby guaranteeing a fair and controlled comparison across different approaches.

## E.2. Hyperparameters Setting In Few-shot Experiments

For the node classification task, we use a fixed set of hyperparameters across all experiments. Specifically, the number of geometric bases $K$ is set to 16, $M$ to 32, and the number of layers to 50. The learning rate is 0.01, with temperature $\tau = 0.3$, and trade-off coefficients $\alpha = 2$ and $\beta = 0.05$. We set the diversity margin to 10, train the model for 60 epochs, and use a batch size of 32. The random seed was fixed to 42 across all experiments.

For graph classification tasks on COX2, NCI1, BZR, COLLAB, IMDB-BINARY, and COLORS-3, most hyperparameters follow the same configuration as the node classification task. The main differences lie in $M$, which is set to 30 for COX2, NCI1, BZR, and COLORS-3, 20 for COLLAB, and 6 for IMDB-BINARY. In addition, the diversity margin is adjusted to 8 for COLLAB and 3 for IMDB-BINARY, while all other hyperparameters remain unchanged.

## E.3. Hardware and Software

We conduct all experiments on the following configurations:

- **Operating System**: Ubuntu 22.04 LTS.

- **CPU**: 13th Gen Intel(R) Core(TM) i9-13900K.

- **GPU**: NVIDIA GeForce RTX 3090 with 24GB of memory.

- **Software**: CUDA 12.1, Python 3.10.19, Pytorch[2] 2.5.1, Pytorch Geometric[3] 2.7.0

# F. Additional Experiments

## F.1. Hyperparameter Sensitivity Analysis

To evaluate the stability of SCGFM, we conducted extensive sensitivity analysis on five key hyperparameters using the PROTEINS dataset under the 2-way 5-shot setting. The results, illustrated in Figure 10, demonstrate that our method maintains high performance across a wide range of configurations.

**Optimization & Regularization ($\tau, \alpha, \beta$):**

---

[2]https://github.com/pytorch/pytorch
[3]https://github.com/pyg-team/pytorch_geometric

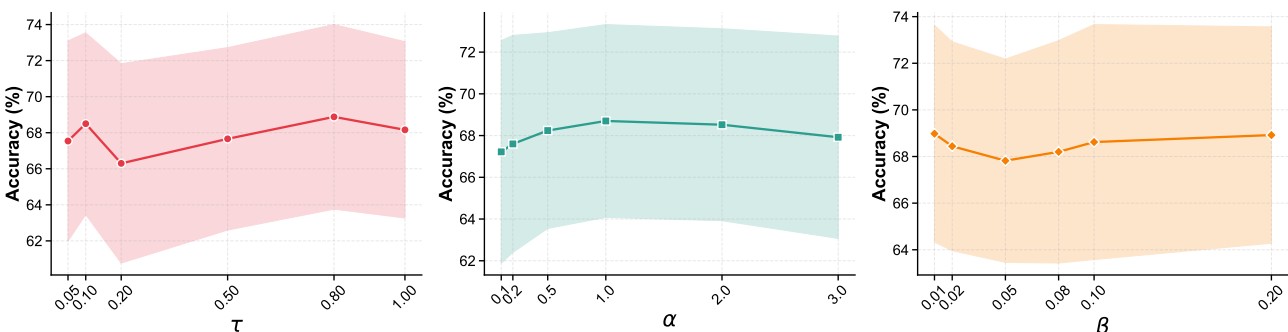

*Figure 10.* Sensitivity analysis of key hyperparameters on PROTEINS classification accuracy. The model exhibits strong robustness, with performance remaining stable across broad ranges of parameter choices.

- **Temperature ($\tau$):** This parameter controls the sharpness of the attention weights in the bases matching step (Eq. 8). The plot indicates an optimal range around $\tau \in [0.5, 0.8]$. Extreme sharpness ($\tau \to 0$) or excessive smoothness ($\tau \to 1.0$) slightly degrades performance, but the overall variance is contained within $2\%$, indicating the key matching mechanism is robust.

- **Loss Coefficients ($\alpha$ and $\beta$):** We denote the weighting coefficients for reconstruction loss and diversity regularization as $\alpha$ and $\beta$, respectively.

  - $\alpha$ shows a "sweet spot" around 1.0, balancing the structural separation (GW loss) and graph-level statistics reconstruction.
  - $\beta$ (diversity weight) exhibits a very flat trend, maintaining accuracy between $68\%$ and $69\%$ across orders of magnitude (0.01 to 0.2). This confirms that while diversity is helpful for preventing mode collapse, the model is not hypersensitive to the exact strength of this regularization.

**Conclusion.**  Overall, SCGFM demonstrates **strong robustness**. The method does not require meticulous hyperparameter tuning to achieve SOTA-level performance, making it highly practical for diverse real-world graph tasks.

### F.2. Additional Ablation Study

In this section, we provide a granular analysis of the ablation study regarding the final graph representation $\mathbf{z}(G)$. According to Eq. 16, the representation is composed of three distinct parts: the geometric coordinate vector $\mathbf{w}$ (derived from OT-based projection), its non-linear projection $f(\mathbf{w})$, and the **structure-aligned feature representation** $\text{vec}(\mathbf{H})$ (derived from the structure-aware re-encoding in Eq. 15). Figure 11 reveal a fundamental dichotomy in graph representation learning, demonstrating how SCGFM effectively disentangles geometry from semantics.

#### F.2.1. THE "FEATURE-DOMINAN" REGIME: THE CASE OF CORA

Cora is a citation network where classification relies heavily on document content (node features).

- **Failure of Pure Geometry ($\mathbf{w}$):** As observed in the "Only $\mathbf{w}$" variant, relying solely on geometric coefficients results in a catastrophic performance drop (Accuracy $\approx 25\%$). This is expected because $\mathbf{w}$ only captures the topological similarity to the bases. Since the citation graphs of different topics often share isomorphic structures (e.g., star-like patterns), the geometric coordinates alone cannot distinguish between classes without semantic content.

- **Dominance of Aligned Features ($\text{vec}(\mathbf{H})$):** The "Only $\text{vec}(\mathbf{H})$" variant achieves $70.55\%$, matching the Full model. Here, $\text{vec}(\mathbf{H})$ represents the input node features (Bag-of-Words) aggregated onto the geometric bases via the transport plan $\mathbf{T}$. This result confirms that preserving the feature content is paramount for citation networks, even after projecting them into the shared latent space.

- **Role of SCGFM:** The Full model successfully identifies that the semantic signal in $\text{vec}(\mathbf{H})$ is the primary discriminator, while implicitly suppressing the noise from the ambiguous geometric signal $\mathbf{w}$ for this specific task.

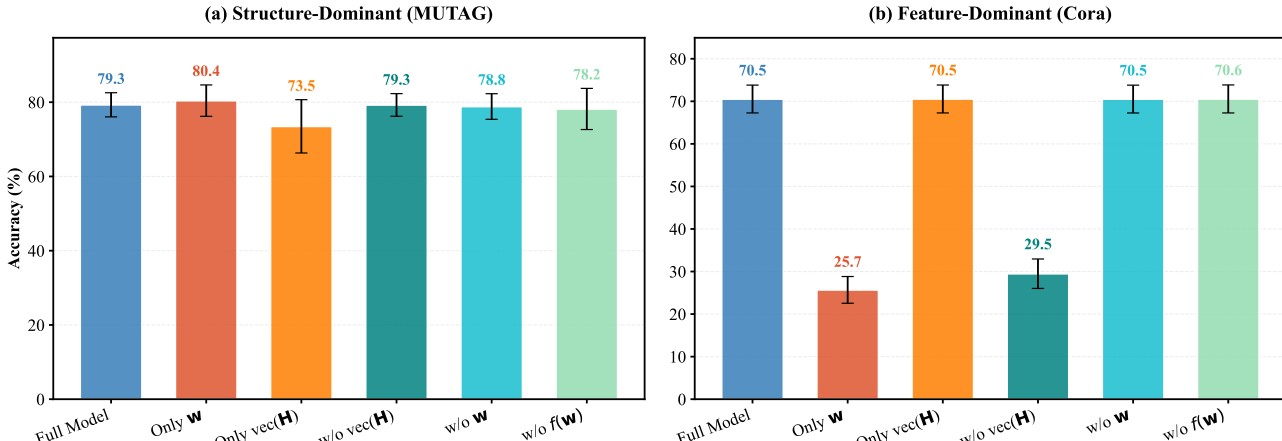

*Figure 11.* Ablation study on the impact of structural and semantic components. We evaluate the model variants on two representative datasets: (a) MUTAG (structure-dominant) and (b) Cora (feature-dominant). $\mathbf{w}$ denotes the learnable structure-related parameter, and $\text{vec}(\mathbf{H}(G))$ denotes the semantic feature vectors. Standard deviation is indicated by error bars. The results show that the SCGFM effectively leverages the dominant modality in each scenario, whereas decoupling the components reveals the dataset-specific reliance on structure versus semantics.

### F.2.2. THE "STRUCTURE-DOMINAN" REGIME: THE CASE OF MUTAG

MUTAG is a molecular dataset where the task is to predict mutagenicity, a property inherently tied to molecular topology (e.g., rings, connectivity).

- **Limitation of Pure Features:** The "Only $\text{vec}(\mathbf{H})$" variant achieves 73.50%. Although these features are structurally aligned, they primarily encode atom types (e.g., C, N, O) mapped to bases. In molecular tasks, the *existence* of a specific pattern (captured by $\mathbf{w}$) is often more critical than the *content* at that pattern (captured by $\text{vec}(\mathbf{H})$).

- **Superiority of Geometric Coordinates ($\mathbf{w}$):** The "Only $\mathbf{w}$" variant achieves a remarkable 80.42%, significantly outperforming the feature-only baseline. This implies that the transport cost vector $\mathbf{w}$—which measures how much effort is needed to morph the input graph into each geometric base—acts as a powerful topological descriptor (fingerprint) for molecular properties.

### F.2.3. THE ROLE OF NON-LINEAR PROJECTION $f(\mathbf{w})$

A natural question arises regarding the necessity of the projection head $f(\cdot)$ in Eq. 16, given that the raw coordinates $\mathbf{w}$ already contain strong structural signals. Based on Table 11, we analyze its function:

- **Latent Space Adaptation:** The geometric bases count $K$ is typically small (e.g., $K = 16$) to ensure orthogonality, whereas the semantic vector $\text{vec}(\mathbf{H})$ usually has a much higher dimensionality. Simply concatenating a low-dim $\mathbf{w}$ with a high-dim $\text{vec}(\mathbf{H})$ would cause the geometric signal to be overwhelmed. $f(\cdot)$ lifts $\mathbf{w}$ to a compatible dimension $(r)$, balancing the contribution of geometry and semantics in the final embedding $\mathbf{z}(G)$.

- **Non-Linear Refinement:** While $\mathbf{w}$ represents linear transport costs, $f(\mathbf{w})$ (parameterized as an MLP) captures non-linear interactions between these costs. Comparing the *Full Model* (79.30%) with *w/o* $f(\mathbf{w})$ (78.18%) on MUTAG, we observe a consistent performance gain ($+1.12\%$). This indicates that the non-linear transformation extracts specific geometric combinations critical for classification that are not present in the raw coordinates.

- **Redundancy vs. Robustness:** Interestingly, on some datasets, $\mathbf{w}$ and $f(\mathbf{w})$ appear redundant (e.g., purely structural performance is similar). However, retaining $f(\mathbf{w})$ ensures the model's robustness. It acts as a buffer that allows the network to learn deeper geometric semantics when raw coordinates are insufficient, without hurting performance in simpler cases.

F.2.4. CONCLUSION: GEOMETRIC AND SEMANTIC COMPLEMENTARITY

The study validates the design of Eq. 16:

1. $w$ answers **"Where does the graph lie in the geometric manifold?"** (crucial for Structure-heavy tasks).

2. $vec(\mathbf{H})$ answers **"What content does the graph carry at those coordinates?"** (crucial for Feature-heavy tasks).

By concatenating them, SCGFM avoids the trade-off inherent in single-view models, achieving robust performance across both regimes.

## F.3. Versatility and Robustness of Representations across Domains

To verify that the structural representations learned by SCGFM are versatile and not strictly tied to a specific metric-based classifier, we extend our evaluation to both **Graph Classification** (PROTEINS) and **Node Classification** (Cora).

We freeze the SCGFM encoder and evaluate five distinct classifiers. This setup tests whether the embeddings contain sufficient information to support different downstream decision mechanisms (metric-based vs. boundary-based).

*Table 10.* Classifier performance comparison across domains. **Left:** Graph Classification on PROTEINS. **Right:** Node Classification on Cora (High-Dim features $\approx$ 28k). PN demonstrates consistent robustness.

| Method | PROTEINS (Graph) | | Cora (Node, High-Dim) | |
|---|---|---|---|---|
| | Accuracy (%) | Std ($\pm$) | Accuracy (%) | Std ($\pm$) |
| PN | **68.46** | 5.15 | **70.55** | 3.27 |
| Random Forest | 66.08 | 5.87 | 70.37 | 3.45 |
| MLP | 64.16 | 6.43 | 65.99 | 4.00 |
| $k$-NN | 65.04 | 7.85 | 61.41 | 4.91 |
| SVM | 64.60 | 5.74 | 59.40 | 5.02 |

**Analysis of Results.** Table 10 reveals two key insights regarding the quality of the learned representations:

**1. Robustness to High Dimensionality (PN).** On the Cora dataset, where the feature dimension expands to approx. 28k, distance-sensitive methods like $k$-NN and SVM suffer from the curse of dimensionality ($61.41\%$ and $59.40\%$, respectively). However, PN adapts well ($70.55\%$) by averaging support samples into prototypes, effectively suppressing high-dimensional noise. This confirms that the SCGFM embeddings maintain a stable global geometry even in extreme high-dimensional settings.

**2. Versatility Beyond Metrics (Random Forest).** Crucially, **Random Forest (RF)** achieves performance nearly identical to PN on Cora ($70.37\%$ vs. $70.55\%$). Unlike PN, RF does not rely on Euclidean distance but on feature splitting. This high performance proves that SCGFM embeddings are **discriminative independent of the distance metric**. The encoder successfully disentangles complex structural information into individual feature dimensions, allowing non-metric classifiers to extract meaningful patterns easily.

**Conclusion.** The experiment confirms that SCGFM produces **task-agnostic and versatile embeddings**. While PN is the most robust choice for few-shot metric learning tasks, the strong performance of Random Forest indicates that the learned representations are rich in semantic content and can be effectively utilized by a broad range of downstream applications, largely unaffected by classifier-specific constraints.

## F.4. Evaluation Beyond ProtoNet-based Classification

To examine whether SCGFM learns generally useful graph representations beyond ProtoNet-based few-shot classification, we evaluate the frozen embeddings on two additional tasks: supervised adaptation with a lightweight MLP classifier. The encoder is pretrained on PROTEINS and transferred to MUTAG and COX2.

**Supervised adaptation.** We freeze the pretrained encoder and train an MLP classifier with limited labeled data on the target dataset. The results are shown in Table 11. SCGFM achieves competitive or better performance than standard GNN baselines (GCN, GIN, GAT) under the same low-label setting.

*Table 11.* Supervised adaptation with a frozen encoder and an MLP classifier.

| Dataset | Labels | SCGFM | Best GNN |
|---------|--------|-------|----------|
| MUTAG | 10% | **83.78** | 81.58 (GIN) |
| MUTAG | 20% | **83.78** | 78.95 (GIN) |
| COX2 | 10% | 80.65 | **81.91** (GCN) |
| COX2 | 20% | **87.10** | 81.91 (GCN) |

## F.5. Additional Scalability Analysis

As illustrated in Figure 6, we conducted comparative experiments on synthetic datasets with a fixed batch of 1,000 graphs, while varying the number of nodes per graph. SCGFM demonstrates superior stability as it scales to massive graphs. The raw data are provided below:

*Table 12.* Runtime comparison on synthetic graphs with increasing graph sizes.

| Node Number | Average Edges | GAT | GIN | GIT | RiemannGFM | SCGFM |
|-------------|---------------|-----|-----|-----|------------|-------|
| 0.1M | 494.67 | 0.12s | 0.10s | 0.21s | 0.24s | 0.84s |
| 0.5M | 12468.16 | 0.70s | 0.19s | 1.15s | 0.54s | 0.89s |
| 1.0M | 49954.51 | 2.61s | 0.61s | 3.73s | 1.46s | 1.47s |
| 3.0M | 449841.264 | OOM | 4.81s | OOM | 8.46s | 7.69s |
| 5.0M | 1249773.292 | OOM | OOM | OOM | OOM | **19.96s** |

These results indicate that SCGFM incurs a slightly higher constant overhead on small graphs due to metric alignment. However, its runtime scales much more gracefully with increasing graph size, and it is the only method that remains executable on 5.0M nodes within a 24 GB VRAM budget. In contrast, all other advanced baselines encounter out-of-memory failures before reaching this scale.

## F.6. Additional Cross-Domain GFM Baselines

To further evaluate the cross-domain transferability of SCGFM, we include three recent GFM or graph pretraining baselines that are designed for cross-domain or heterogeneous graph scenarios: SAMGPT, BRIDGE, and MDGFM. All methods are evaluated under the same 5-shot cross-domain graph classification setting.

Since these methods adopt different downstream adaptation strategies, we report two complementary settings for fair comparison. The *native* setting preserves each method's original target-side prompt or adaptation mechanism whenever supported by the released code. The *aligned* setting freezes the pretrained encoder as much as possible, minimizes target-side adaptation, and evaluates all methods using ProtoNet under the same fixed-encoder protocol as SCGFM.

*Table 13.* Additional 5-shot cross-domain graph classification results with recent GFM and graph pretraining baselines.

| Model | BZR → PROTEINS (%) | PROTEINS → BZR (%) |
|-------|--------------------|--------------------|
| BRIDGE (native) | 51.06 ± 5.80 | 56.38 ± 6.89 |
| SAMGPT (native) | 51.36 ± 5.52 | 57.90 ± 7.47 |
| MDGFM (native) | 57.06 ± 7.70 | 53.00 ± 6.02 |
| BRIDGE (aligned) | 49.94 ± 0.42 | 51.24 ± 5.94 |
| SAMGPT (aligned) | 50.16 ± 0.89 | 53.12 ± 7.08 |
| MDGFM (aligned) | 54.04 ± 6.78 | 51.94 ± 5.30 |
| SCGFM | **68.32 ± 4.65** | **58.60 ± 6.75** |

As shown in Table 13, SCGFM achieves the best performance in both transfer directions. It also outperforms the strongest native baseline, demonstrating its stronger robustness under heterogeneous cross-domain transfer.

## F.7. Baseline Selection and Evaluation Protocol

We clarify the rationale behind our baseline selection and evaluation protocol. Recent GFMs such as OFA, SAMGPT, BRIDGE, MDGFM, and RAG4GFM are important related methods, but many of them follow an *interface-centric* paradigm, where prompts, instructions, retrieval modules, or task-specific target-side adaptation are used to bridge different graph

domains. In contrast, SCGFM is designed as a *structure-centric* model: it constructs a shared geometric coordinate system directly from graph topology and transfers through structure-aligned representations, without relying on textual semantics or prompt-based interfaces.

This distinction also affects the evaluation protocol. In the main paper, we evaluate SCGFM under a strict *fixed-encoder few-shot* setting, where the pretrained model is frozen and only a lightweight downstream classifier is used. This protocol is intended to measure the intrinsic transferability and representation quality of the pretrained encoder. Prompt-based GFMs, however, are often designed to exploit target-side prompts, adapters, or retrieval-based augmentation during transfer. Directly comparing their native target-adaptation pipelines with a frozen SCGFM encoder would therefore conflate representation quality with additional adaptation mechanisms.

To address this issue, we distinguish between two complementary settings in the supplementary comparison (Table 13). These comparisons complement the main results and show that SCGFM remains competitive, and in most cases superior, under both native and protocol-aligned cross-domain evaluation settings.

## G. Discussion

### G.1. Representation Components and Structure-conditioned Feature Projection

SCGFM uses a combined representation

$$Z(G) = [\mathbf{w}, f(\mathbf{w}), \text{vec}(\mathbf{H})],$$

where $\mathbf{w}$ denotes the structural coordinates, $f(\mathbf{w})$ denotes the decoder output, and $\text{vec}(\mathbf{H})$ denotes the vectorized structure-conditioned feature representation. This design allows the representation to adapt to both structure-dominant and feature-dominant regimes.

Importantly, $\text{vec}(\mathbf{H})$ is not an arbitrary high-dimensional lift of node features. It is conditioned on graph structure through the OT coupling $\mathbf{T}$ in Eq. 15, which aligns the input graph with the learned geometric bases. Thus, $\text{vec}(\mathbf{H})$ reflects the feature information after structural alignment.

To verify this, we conduct a diagnostic experiment on 1000 Cora ego-graphs. We fix the node feature matrix $\mathbf{X}$ and perturb only the graph topology $\mathbf{A}$ by rewiring edges with probability $\epsilon$. We then measure the cosine distance of the representation before and after rewiring. As a same-dimensional structure-agnostic control, we use `pool_tiled`, which preserves dimensionality but removes OT-based structural conditioning. The results are shown in Table 14.

*Table 14.* Diagnostic study with fixed node features and rewired graph structure on 1000 Cora ego-graphs. $\Delta$ denotes cosine distance before and after rewiring.

| Rewire $\epsilon$ | $\Delta\text{vec}(\mathbf{H})$ Full | $\Delta\text{vec}(\mathbf{H})$ `pool_tiled` | $\Delta\mathbf{w}$ Full |
|---|---|---|---|
| 0.00 | $1.0\times10^{-8} \pm 7.0\times10^{-8}$ | $1.0\times10^{-8} \pm 8.0\times10^{-8}$ | $\sim 0$ |
| 0.30 | $1.86\times10^{-2} \pm 3.25\times10^{-2}$ | $1.0\times10^{-8} \pm 8.0\times10^{-8}$ | $7.9\times10^{-7} \pm 8.9\times10^{-7}$ |
| 0.70 | $5.26\times10^{-2} \pm 7.04\times10^{-2}$ | $1.0\times10^{-8} \pm 8.0\times10^{-8}$ | $2.2\times10^{-6} \pm 2.3\times10^{-6}$ |

The full model's $\text{vec}(\mathbf{H})$ changes monotonically with structural perturbation, whereas the same-dimensional structure-agnostic control remains nearly unchanged. This confirms that the contribution of $\text{vec}(\mathbf{H})$ comes from structure-conditioned alignment rather than dimensionality alone.

### G.2. Support-set Criterion for Component Selection

Different datasets may benefit from different representation components. To provide a practical criterion for component selection, we compute the within-class scatter $S_w$ and between-class scatter $S_b$ of each component on the few-shot support set using the frozen encoder. We define a Fisher-style separability ratio as

$$R = \frac{S_b}{S_w}. \tag{18}$$

A larger $R$ indicates stronger class separability for the corresponding component.

Table 15 reports the ratios of $\mathbf{w}$ and $\text{vec}(\mathbf{H})$ on Cora, PROTEINS, and MUTAG.

*Table 15.* Fisher-style separability ratio $R = S_b/S_w$ for different representation components.

| Dataset | $R$ of $\mathbf{w}$ / $\mathrm{vec}(\mathbf{H})$ | Dominant Component |
|---|---|---|
| Cora | 58.1 / **231.4** | $\mathrm{vec}(\mathbf{H})$ |
| PROTEINS | **304.3** / 71.7 | $\mathbf{w}$ |
| MUTAG | **143.5** / 105.6 | $\mathbf{w}$ |

This provides a simple practical rule: for a new dataset, the component with the larger support-set ratio $R$ can be assigned higher importance. The criterion also explains the observed behavior across datasets: Cora benefits more from the structure-conditioned feature representation, whereas PROTEINS and MUTAG benefit more from the structural coordinate representation.

### G.3. Quality of the SGW Approximation

SCGFM adopts the Sliced Gromov–Wasserstein distance to approximate the exact GW distance for scalable training. To evaluate the approximation quality, we compute exact GW and SGW distances on 500 graph pairs from PROTEINS. The Pearson correlation between the two distances is

$$\rho = 0.7576, \qquad p < 10^{-90}.$$

This is consistent with the NCI1 result in Figure 9, where the correlation is $\rho = 0.7005$.

These results indicate that SGW preserves the relative geometric ordering of graph pairs while reducing the computational cost from approximately $O(N^3)$ for exact GW to $O(N \log N)$ for SGW-based computation. This reduction is crucial for scalable pretraining across diverse graph datasets.

### G.4. Effect of the Linear Reconstruction Surrogate

Exact GW barycenter computation requires nested OT iterations, which are computationally expensive and difficult to integrate into end-to-end gradient-based training. SCGFM therefore uses a linear reconstruction surrogate as defined in Eq. 7, enabling efficient differentiable reconstruction from structural coordinates.

We conduct a diagnostic experiment on PROTEINS over 200 epochs. Table 16 compares the exact barycenter reconstruction and the linear surrogate.

*Table 16.* Comparison between exact GW barycenter reconstruction and the linear surrogate on PROTEINS.

| Metric | Exact Barycenter | Linear Surrogate |
|---|---|---|
| Avg. SGW recon. error | 0.018739 | 0.018776 (+0.000037) |
| Inference time / graph | 1.47 ms | **0.04 ms** ($36\times$) |
| End-to-end trainable | No | **Yes** |

The linear surrogate yields almost identical reconstruction fidelity, with only a negligible increase in SGW reconstruction error. Meanwhile, it reduces inference time from 1.47 ms to 0.04 ms per graph and supports end-to-end training. Therefore, the surrogate provides a practical and scalable approximation to GW barycentric reconstruction.

### G.5. Sensitivity to Sampling Strategy and Subgraph Size

For node-level transfer, each target node is represented by a sampled ego-subgraph. We study the sensitivity of SCGFM to both the sampling strategy and the sampled subgraph size on the CiteSeer $\rightarrow$ Cora 5-shot transfer setting. We compare $k$-hop sampling, random walk (RW) sampling, and personalized PageRank (PPR) sampling.

Table 17 reports the results under different subgraph sizes. Performance generally improves when the subgraph size increases from 16 to 64 and then saturates around 64–128 nodes.

We further compare sampling strategies across different models. For each model and strategy, Table 18 reports the best accuracy over the tested subgraph sizes.

The results suggest two observations. First, SCGFM benefits from moderately sized subgraphs, with performance saturating

*Table 17.* Sensitivity of SCGFM to sampling strategy and subgraph size on CiteSeer → Cora under the 5-shot setting.

| Strategy | Size = 16 | Size = 64 | Size = 128 |
|---|---|---|---|
| $k$-hop | $62.97 \pm 3.70$ | $67.06 \pm 3.05$ | $65.93 \pm 2.82$ |
| RW | $59.17 \pm 3.98$ | $66.84 \pm 3.87$ | $69.67 \pm 2.87$ |
| PPR | $\mathbf{64.73} \pm 3.53$ | $\mathbf{71.23} \pm 2.73$ | $\mathbf{71.39} \pm 2.55$ |

*Table 18.* Comparison of sampling strategies across models on CiteSeer → Cora. Each entry reports the best accuracy over tested subgraph sizes.

| Model | $k$-hop | RW | PPR |
|---|---|---|---|
| SCGFM | 67.34@96 | 69.67@128 | **71.39@128** |
| GCN | **45.53@128** | 38.99@128 | 44.44@128 |
| GIN | **45.86@96** | 38.73@128 | 42.87@128 |

around 64–128 nodes. Second, the preferred sampling strategy is model-dependent. SCGFM achieves the best performance with PPR sampling, which better preserves globally relevant structural context, while GCN and GIN obtain their best results with $k$-hop neighborhoods.

## H. Proof

### H.1. Proof of Theorem 3.2

*Proof.* Let $G$ and $G'$ be two input graphs with structural distance $d_{GW}(G, G')$.

**Stability of Structural Coordinates (w)**

The coordinates are derived from Softmax-normalized distances to fixed bases. It has been established that the Softmax function is Lipschitz continuous with a constant of $L_{sm}$ with respect to the standard Euclidean norm (Gao & Pavel, 2017). Combined with the scaling factor $1/\tau$, we have:

$$\|\mathbf{w} - \mathbf{w}'\|_2 = \| \operatorname{softmax}(-\mathbf{d}/\tau) - \operatorname{softmax}(-\mathbf{d}'/\tau)\| \tag{19}$$

$$\leq L_{sm} \cdot \| - \mathbf{d}/\tau - (-\mathbf{d}'/\tau)\| \tag{20}$$

$$= \frac{L_{sm}}{\tau} \|\mathbf{d} - \mathbf{d}'\|_2 \tag{21}$$

The vector $\mathbf{d}(G) \in \mathbb{R}^K$ consists of the GW distances from graph $G$ to the set of fixed geometric bases $\mathcal{B} = \{B_1, B_2, \ldots, B_K\}$, i.e., $d_k = d_{GW}(G, B_k)$.

Since the GW distance $d_{GW}$ satisfies the triangle inequality on the considered graph space, it also satisfies the reverse triangle inequality:

$$|d_{GW}(G, B_k) - d_{GW}(G', B_k)| \leq d_{GW}(G, G') \tag{22}$$

for any basis $B_k$.

Applying this inequality component-wise to the $l_2$-norm formulation of $\|\mathbf{d} - \mathbf{d}'\|_2$, we obtain:

$$\|\mathbf{d} - \mathbf{d}'\|_2 = \sqrt{\sum_{k=1}^{K} (d_{GW}(G, B_k) - d_{GW}(G', B_k))^2} \tag{23}$$

$$\leq \sqrt{\sum_{k=1}^{K} (d_{GW}(G, G'))^2} \tag{24}$$

$$= \sqrt{K(d_{GW}(G, G'))^2} \tag{25}$$

$$= \sqrt{K} \cdot d_{GW}(G, G') \tag{26}$$

Combining this result with Eq. 21, we obtain the final Lipschitz bound for the weight vector $\mathbf{w}$:

$$\|\mathbf{w} - \mathbf{w}'\|_2 \leq \frac{L_{sm}}{\tau} \|\mathbf{d} - \mathbf{d}'\|_2 \tag{27}$$

$$\leq \frac{\sqrt{K}L_{sm}}{\tau} \cdot d_{GW}(G, G') \tag{28}$$

This confirms that the structural coordinates representation $\mathbf{w}$ is Lipschitz continuous with respect to the input graph structure, with a Lipschitz constant $L_{\mathbf{w}} = \frac{\sqrt{K}L_{sm}}{\tau}$.

$\square$

