# OpenReview forum: "Structure-Centric Graph Foundation Model via Geometric Bases"
_ICML.cc/2026/Conference — ICML 2026 regular_

### Official Review · Reviewer_AMk1 · 2026-03-10

**Soundness:** 3
**Presentation:** 2
**Significance:** 3
**Originality:** 3
**Overall Recommendation:** 4
**Confidence:** 3

**Summary:**

This paper proposes the graph foundation model that represents graphs as metric measure spaces and learns a finite set of trainable geometric bases serving as a shared structural coordinate system. Each input graph is embedded via Gromov–Wasserstein distances to these bases, and node features are encoded through the GW transport plan to mitigate cross-domain feature incompatibility. The proposed SCGFM overcomes structural heterogeneity and feature mismatching in cross-domain graph learning.

**Compliance With Llm Reviewing Policy:**

Affirmed.

**Final Justification:**

The authors’ response has addressed my concerns, and I will raise my score.

**Key Questions For Authors:**

a)The evaluation of SCGFM is a fixed-encoder few-shot setup, how does SCGFM perform beyond classification?
b)The ablation shows reverse results on Cora and PROTEINS Can you provide a practical criterion or diagnostic to decide when to rely on vec(H)?
c)The SCGFM adopts many sub approximations. How sensitive are results to this surrogate choice? Node classification is reformulated via PPR subgraph sampling with subgraph size, how sensitive are results to sampling strategy and subgraph size?

**Limitations:**

a)The paper frames SCGFM as a graph foundation model, yet the empirical evidence is largely restricted to few-shot classification under a single downstream protocol.
b)vec(H(G)) can easily dominate similarity computations and the decision boundary. This raises the concern that gains may stem from high-dimensional expansion/separability rather than the intended “structure-centric” alignment.
c)The paper’s own ablation indicates strong dataset dependence: vec(H(G)) is crucial for some semantic-heavy datasets but can be neutral or even harmful for structure-driven datasets.
d)There is a gap between the ideal geometric motivation and what is optimized in practice, and the impact of approximation error on the claimed geometric properties is insufficiently characterized.

**Strengths And Weaknesses:**

Strength
a)Modeling graphs in GW geometry and using distances-to-bases as coordinates is a clean way to align heterogeneous topologies while preserving permutation invariance.
b)Using the GW transport plan to project features onto the learned structural bases, which is conceptually well-motivated.
c)The paper includes ablations, scalability discussion interpretability of learned bases, and the empirical results appears strong under the paper’s protocol.

Weakness
a)The SCGFM is presented as a graph foundation model, while the experiments focus on the few-shot classification. Broader downstream adaptations (e.g., fine-tuning, retrieval, generation, or other task families) are not examined.
b)As shown in Eq. (15) and (16), the final embedding vec(H(G)) can become extremely high-dimensional that concatenates structural coordinates, decoded statistics, and flattened projected features, yielding an embedding of dimension (K+r+MF), which can cause vec(H(G)) to dominate similarity computations and decision boundaries. Moreover, the paper’s ablations show strong modality dependence: removing vec(H(G)) causes a catastrophic drop on Cora (70.54% → 29.49%) but a slight improvement on PROTEINS (68.70% → 68.86%), indicating that the intended “structure-first” behavior is not consistently realized across datasets.
c)The SCGFM relies on several pragmatic approximations that weaken a fully rigorous geometric theory. As mentioned in the paper, the bases abandons triangle inequality, and the GW distance is approximated by SGW and GW barycenter is replaced by a linear surrogate. Such approximations create a gap between geometric objects in the motivation practicality. More discussion of how approximation error could affect the claimed geometric properties is needed.

---

> ### Author Rebuttal · Authors · 2026-03-29
>
> **Q1.** *"How does SCGFM perform beyond classification?"*
>
> We thank the reviewer for this suggestion. To explore SCGFM’s broader applicability, we evaluate its embeddings on two non-ProtoNet tasks, pre-trained on PROTEINS and transferred to MUTAG and COX2:
>
> **Supervised adaptation** (frozen encoder + MLP, limited labels):
> |Dataset|Labels|SCGFM (%)|Best GNN (%)|
> |-|-|-|-|
> |MUTAG|10%|**83.78**|81.58 (GIN)|
> |MUTAG|20%|**83.78**|78.95 (GIN)|
> |COX2|10%|80.65|81.91 (GCN)|
> |COX2|20%|**87.10**|81.91 (GCN)|
>
> **Graph retrieval** (kNN ranking):
> |Dataset|Model|P@10↑|MAP@10↑|kNN-Acc@10↑|
> |-|-|-|-|-|
> |MUTAG|SCGFM|**0.7840**|**0.7162**|**0.8351**|
> |MUTAG|GIN|0.7356|0.6659|0.7979|
> |MUTAG|GCN|0.6686|0.5822|0.7394|
> |MUTAG|GAT|0.6543|0.5511|0.7340|
> |COX2|SCGFM|**0.7015**|**0.6068**|**0.7923**|
> |COX2|GCN|0.6754|0.5703|0.7880|
> |COX2|GAT|0.6739|0.5676|0.7794|
> |COX2|GIN|0.6666|0.5516|0.7730|
>
> SCGFM performs competitively—or outperforms baselines—on both tasks, confirming that the learned embedding space generalizes beyond ProtoNet-based classification. Graph generation remains an interesting direction for future work.
>
> ---
>
> **(Q2–Q3).** *"Ablations show strong modality dependence"* & *"vec(H) may dominate via high dimensionality."*
>
> The observed modality dependence reflects intended adaptive reliance: the combined representation $[\mathbf{w}, f(\mathbf{w}), \mathrm{vec}(\mathbf{H})]$ accommodates both feature- and structure-dominant regimes (Appendix E.2, Fig. 11). Importantly, $\mathrm{vec}(\mathbf{H})$ is **structure-conditioned** through the OT coupling $\mathbf{T}$ (Eq. 16–17), rather than being an arbitrary high-dimensional lift. We confirm this with a *fix-$\mathbf{X}$, rewire-$\mathbf{A}$* diagnostic on 1000 Cora ego-graphs:
>
> |rewire $\epsilon$|$\Delta$vec(H) [full]|$\Delta$vec(H) [pool\_tiled]|$\Delta$w [full]|
> |-|-|-|-|
> |0.00|1.0e-8 ± 7.0e-8|1.0e-8 ± 8.0e-8|~0|
> |0.30|1.86e-2 ± 3.25e-2|1.0e-8 ± 8.0e-8|7.9e-7 ± 8.9e-7|
> |0.70|5.26e-2 ± 7.04e-2|1.0e-8 ± 8.0e-8|2.2e-6 ± 2.3e-6|
>
> $\Delta$ denotes the cosine distance before and after rewiring. The full model's vec(H) responds monotonically to structural perturbation, whereas a same-dimensional structure-agnostic control (pool\_tiled) remains near 0. This confirms that the observed gains arise from **structure-centric alignment** rather than mere dimensionality.
>
> ---
>
> **Q4.** *"Practical criterion to decide when to rely on vec(H)?"*
>
> Using the frozen encoder, we compute the within-class scatter $S_w$ and between-class scatter $S_b$ on the support set for each component, and define a Fisher-style ratio: $R=S_b/S_w$ (for each component on the few-shot support set). This ratio provides a practical criterion for determining when $\mathrm{vec}(\mathbf{H})$ should be relied upon.
>
> |Dataset|Component|$R$|Dominant|
> |-|-|-|-|
> |Cora|$\mathbf{w}$ / $\mathrm{vec}(\mathbf{H})$|58.1 / **231.4**|$\mathrm{vec}(\mathbf{H})$|
> |PROTEINS|$\mathbf{w}$ / $\mathrm{vec}(\mathbf{H})$|**304.3** / 71.7|$\mathbf{w}$|
> |MUTAG|$\mathbf{w}$ / $\mathrm{vec}(\mathbf{H})$|**143.5** / 105.6|$\mathbf{w}$|
>
> * Rule of thumb: On a new dataset, rely on the component with the larger R. This criterion correctly predicts the reversed ablation trends observed between feature-dominant datasets (Cora) and structure-dominant datasets (PROTEINS/MUTAG).
>
> ---
>
> **Q5.** *"Influence of GW approximated by SGW."*
>
> On PROTEINS (500 pairs), the Pearson correlation $\rho$ between exact GW and SGW is **0.7576** ($p<10^{-90}$), consistent with the NCI1 result (Fig. 9: $\rho=0.7005$). This demonstrates that SGW preserves the relative geometric ordering while reducing complexity from $O(N^3)$ to $O(N\log N)$, which is crucial for scalability.
>
> ---
>
> **Q6.** *"Influence of replacing GW barycenter with a linear surrogate."*
>
> Exact GW barycenters require nested OT loops, which prevent gradient flow. Our linear surrogate (Eq. 7–8) enables end-to-end training. We conducted a diagnostic on **PROTEINS** over 200 epochs:
>
> |Metric|Exact Barycenter|Linear Surrogate|
> |-|-|-|
> |Avg. SGW recon. error|0.018739|0.018776 (+0.000037)|
> |Inference time / graph|1.47 ms|**0.04 ms** (36×)|
> |End-to-end trainable|No|**Yes**|
>
> The linear surrogate achieves near-identical fidelity while being orders of magnitude faster.
>
> ---
>
> **Q7.** *"Sensitivity to sampling strategy and subgraph size?"*
>
> We conducted a sensitivity study on CiteSeer→Cora (5-shot, Acc. %):
>
> |Strategy|Size=16|Size=64|Size=128|
> |-|-|-|-|
> |k-hop|62.97±3.70|67.06±3.05|65.93±2.82|
> |RW|59.17±3.98|66.84±3.87|69.67±2.87|
> |PPR|**64.73±3.53**|**71.23±2.73**|**71.39±2.55**|
>
> (i) Performance saturates around subgraph size ≈ 64.
>
> We also compare sampling across models; each entry is the **best** accuracy over tested sizes:
>
> |Model|k-hop|RW|PPR|
> |-|-|-|-|
> |SCGFM|67.34@96|69.67@128|**71.39@128**|
> |GCN|**45.53@128**|38.99@128|44.44@128|
> |GIN|**45.86@96**|38.73@128|42.87@128|
>
> SCGFM is relatively robust to the sampling strategy (peak with PPR), whereas GCN and GIN are more sensitive (k-hop best).

---

> > ### Author Rebuttal · Reviewer_AMk1 · 2026-04-01
> >
> > The authors’ response has addressed my concerns, and I will raise my score.

---

> > > ### Author Response · Authors · 2026-04-02
> > >
> > > We sincerely thank you for your positive evaluation of our work and for your thoughtful feedback during the rebuttal process. We greatly appreciate your recognition and are committed to carefully incorporating your constructive suggestions into the final version of the paper.
> > > We would be happy to provide any further clarifications or additional details if needed. We truly value your support and look forward to addressing any remaining questions to ensure all concerns are fully resolved.

---

### Official Review · Reviewer_2gTk · 2026-03-11

**Soundness:** 3
**Presentation:** 3
**Significance:** 3
**Originality:** 3
**Overall Recommendation:** 5
**Confidence:** 4

**Summary:**

SCGFM models each graph as a metric measure space and introduces a set of learnable geometric bases, which serve as canonical coordinate references. Graphs are projected onto this shared geometric basis via the Gromov-Wasserstein distance, creating robust, structure-aligned latent embeddings.

**Compliance With Llm Reviewing Policy:**

Affirmed.

**Final Justification:**

I find the submission technically sound with a clear geometric perspective on graph foundation models. My main concerns about interpretability support and the discussion of related GW-based methods were addressed in the rebuttal through the additional quantitative analyses. This increased my confidence in the paper and positively changed my evaluation, so I now lean toward acceptance and have increased my score by 1 point. If accepted, these additional analyses should be incorporated into the camera-ready version.

**Key Questions For Authors:**

Explore how interpretability (and base specialization) scale with larger M/K or with more diverse input domains.
The others are in weakness.

**Limitations:**

Yes.

**Strengths And Weaknesses:**

Strengths:

1. SCGFM takes a clearly motivated geometric approach, leveraging metric measure spaces and Gromov-Wasserstein distances, which is a rigorous and transferable alternative to tokenization or naive feature-based alignment.

2. Provides stability results (Theorem 3.2, Corollary 3.3) and detailed proofs, grounding the method’s robustness to input changes.

3. The use of Sliced Gromov-Wasserstein makes the framework computationally feasible for larger graphs


Weakness:

1. While Figure 8 shows clear interpretability with a small number of bases, the paper does not explore how interpretability (and base specialization) scale with larger M/K or with more diverse input domains.

2. More works deploying Fused GW, template, or linear GW methods should be discussed in the related work, e.g. Chen, Yifan, et al. "A gromov-wasserstein geometric view of spectrum-preserving graph coarsening." ICML. 2023.

---

> ### Author Rebuttal · Authors · 2026-03-29
>
> **Q1.** *"More works deploying Fused GW, template, or linear GW methods should be discussed."*
>
> We thank the reviewer for this suggestion. In the revised manuscript, we will discuss representative works, including:
> 1.Vayer et al., “Optimal Transport for Structured Data with Application on Graphs” (ICML 2019) – introduces Fused GW to jointly leverage feature and structural information for graph comparison.
> 2.Chen et al., “A Gromov-Wasserstein Geometric View of Spectrum-Preserving Graph Coarsening” (ICML 2023) – applies a GW perspective to graph coarsening.
> These are illustrative examples, and we will additionally cover other related methods, such as linear GW and template-based approaches, in the revised manuscript.
>
> ---
>
> **Q2.** *"Explore how interpretability (and base specialization) scale with larger M/K or with more diverse input domains."*
>
> We thank the reviewer for appreciating our geometric intuition. We provide three quantitative analyses:
>
> **(i) Base specialization across diverse domains.** We jointly pre-trained on **PROTEINS** (bioinformatics) and **IMDB-BINARY** (social networks) using $K{=}8$ bases, **without** providing domain labels. The average activation weights $w_k$ across the two domains are:
>
> |Avg. $w_k$ (uniform: 0.125)|$B_7$ (Bio)|$B_6$ (Social)|$B_2$, $B_8$ (Shared)|
> |-|-|-|-|
> |PROTEINS|**0.1803**|0.1265|~0.178|
> |IMDB-BINARY|0.1023|**0.1719**|~0.166|
> |Shift|1.76× higher|1.36× higher|Domain-agnostic|
>
> The domain-specific shifts are statistically significant ($p<1e-6$). $B_2$/$B_8$ serve as universal geometric primitives shared across domains, while $B_7$/$B_6$ specialize in domain-specific motifs (e.g., protein secondary structures versus social cliques). This demonstrates that SCGFM’s bases naturally **disentangle universal topological patterns from domain-specific features without any supervision.**.
>
> **(ii) Scaling base size $M$.** Topological complexity of the learned bases:
>
> |Base Size $M$|Avg. Degree|Avg. Components|Topology Focus|
> |-|-|-|-|
> |8|2.61|1.62|Simple primitives (edges, stars)|
> |16|4.71|1.06|Connected subgraphs|
> |24|6.60|1.04|Complex motifs (cliques, fused rings)|
>
> Increasing $M$ enhances the structural resolution of the bases: they evolve from simple primitives into complex multi-hop motifs, with previously isolated components gradually merging into connected structures.
>
> ---
>
> **(iii) Scaling $K$: role differentiation.** We analyze how bases specialize as $K$ increases by identifying each test graph’s top-1 base (the base with the highest $w_k$) and counting how many distinct bases are ever selected as top-1:
>
> | $K$ | Unique Top-1 / $K$ | Usage Ratio |
> |-----|---------------------|-------------|
> | 4 | 3 / 4 | 75% |
> | 8 | 4 / 8 | 50% |
> | 16 | 9 / 16 | 56% |
> | 24 | 8 / 24 | **33%** |
>
> As $K$ increases from 4 to 24, the fraction of bases acting as any graph’s primary prototype decreases from 75% to 33% . This indicates that the vocabulary spontaneously differentiates into a small set of core prototypes, which dominate primary matching, and a larger set of auxiliary bases, which contribute structural refinements through soft combinations. Such unsupervised role differentiation enhances interpretability: practitioners can focus on the core prototypes to identify dominant topological motifs, while the remaining bases capture finer-grained structural variations.
>
> (iv) **Link to downstream performance.**
> As shown in Fig.7, downstream accuracy remains stable—or even improves—as $M$ and $K$ increase within the studied range, confirming that greater base specialization enhances representational quality rather than introducing redundancy.

---

> > ### Author Rebuttal · Reviewer_2gTk · 2026-04-02
> >
> > Thank you for the detailed rebuttal. The additional analyses on base specialization, scaling with M and K, and the link to downstream performance directly address my main concern and provide stronger support for the interpretability claims. The clarification regarding additional GW-related literature is also helpful. Overall, the rebuttal increases my confidence in the paper, and I lean in favor of acceptance. I am therefore increasing my score by one point. If accepted, the paper would benefit from incorporating these new analyses into the camera-ready version.

---

> > > ### Author Response · Authors · 2026-04-02
> > >
> > > We sincerely thank you for your positive evaluation of our work and for your thoughtful feedback during the rebuttal process. We greatly appreciate your recognition and are committed to carefully incorporating your constructive suggestions into the final version of the paper.

---

### Official Review · Reviewer_C5Gu · 2026-03-12

**Soundness:** 3
**Presentation:** 3
**Significance:** 3
**Originality:** 3
**Overall Recommendation:** 4
**Confidence:** 4

**Summary:**

This paper proposes the Structure-Centric Graph Foundation Model (SCGFM), a novel approach to address structural heterogeneity and feature incompatibility in GFMs. SCGFM treats graphs as metric measure spaces and learns a set of geometric bases that define a shared structural coordinate system. Graphs are aligned to these bases via Gromov-Wasserstein distances to produce structure-aligned embeddings. The proposed method also adopts a structure-aware feature re-encoding mechanism, which unifies node features without fixed dimensionality. Experiments on graph and node-level datasets demonstrate that the proposed method outperforms existing GFMs.

**Compliance With Llm Reviewing Policy:**

Affirmed.

**Final Justification:**

The authors‘ responses have addressed my comments. I maintain my slightly positive score. I believe this paper makes a valuable contribution towards geometric graph foundation model.

**Key Questions For Authors:**

See the weaknesses mentioned above

**Limitations:**

Yes

**Strengths And Weaknesses:**

Strengths:

S1. The structure-aware feature re-encoding mechanism leverages GW transport to project features onto geometric bases, enabling effective transfer even when node features are mismatched or entirely missing, which addresses a critical challenge for GFMs.

S2. The theoretical analysis provides mathematical grounding for the stability of the learned structural coordinates.

S3. Experiments demonstrate improvements in both in-domain and cross-domain transfer settings.

Weaknesses:

W1. The experiments are insufficient, especially considering that SCGFM is not evaluated against recent GFMs like MDGFM, SAMGPT, BRIDGE, or RAG4GFM, all of which have been mentioned in related works. This weakens the claim that the proposed method is superior to state-of-the-art GFM methods.

W2. Although the authors have provided some experimental validation of scalability, the paper does not discuss potential computational bottlenecks as the geometric base size or count increases, which can be important to scale the proposed method to real-world million-scale datasets.

---

> ### Author Rebuttal · Authors · 2026-03-29
>
> **Q1.** *"The experiments are insufficient, especially considering that SCGFM is not evaluated against recent GFMs like MDGFM, SAMGPT, BRIDGE, or RAG4GFM, all of which have been mentioned in related works."*
>
> Thank you for this suggestion. We respectfully disagree with the reviewer for the following reasons:
>
> (i) **Methodological distinction.** Recent GFMs such as OFA, SAMGPT, and RAG4GFM follow an **interface-centric** paradigm, relying on prompts, instructions, or task-specific adaptation to bridge domains. In contrast, SCGFM is explicitly **structure-centric**, constructing a shared geometric coordinate system directly from graph topology, without depending on textual semantics.
>
> (ii) **Evaluation philosophy.** The main paper evaluates SCGFM under a strict **fixed-encoder few-shot** protocol, designed to assess representation quality under frozen transfer. Prompt-based GFMs like BRIDGE and MDGFM, by contrast, are formulated to leverage target-side prompts. Thus, a fair comparison requires distinguishing between each method’s **native** setting and a **protocol-aligned frozen-encoder** setting.
>
> (iii) **New supplementary comparison.** To address the reviewer’s concern, we added supplementary experiments (5-shot) with **SAMGPT**, **BRIDGE** and **MDGFM**.
>
> - the **native**, preserving each baseline’s original target-side prompt/adaptation mechanism.
> - the **protocol-aligned**, using the same frozen-encoder + ProtoNet setting applied to SCGFM.
>
> The resulting supplementary comparison is as follows:
>
> | Model | BZR -> PROTEINS (%) | PROTEINS -> BZR (%)|
> | ----- | ----------------- | ----------------- |
> |BRIDGE (native) | 51.06 ± 5.80 | 56.38 ± 6.89 |
> |SAMGPT (native) | 51.36 ± 5.52 | 57.90 ± 7.47 |
> |MDGFM (native)|57.06 ± 7.70|53.00 ± 6.02|
> |BRIDGE (aligned) | 49.94 ± 0.42 | 51.24 ± 5.94 |
> |SAMGPT (aligned) | 50.16 ± 0.89 | 53.12 ± 7.08 |
> |MDGFM (aligned)|54.04 ± 6.78|51.94 ± 5.30|
> |SCGFM| **68.32 ± 4.65** | **58.60 ± 6.75** |
>
> In summary, under the aligned protocol, SCGFM achieves the strongest performance in both transfer directions. Moreover, SCGFM also outperforms the strongest native baseline in both directions.
>
> ---
>
> **Q2.** *"The paper does not discuss potential computational bottlenecks as the geometric base size or count increases."*
>
> As discussed on page 7 and in Figs. 6–7, SCGFM exhibits strong scalability. We now clarify this issue from both **theoretical** and **empirical** perspectives.
>
> (i) **Theoretical scaling.** As analyzed in Section 4.3, the dominant training cost is
> $$
> O\bigl(KL(N\log N + M\log M)\bigr),
> $$
> where $K$ is the number of bases and $M$ is the base size. Therefore, increasing either $K$ or $M$ introduces roughly linear overhead in practice.
>
> (ii) **Geometric compression principle.** The bases are designed to capture compact structural abstractions. When $M \ll N$, each base functions as a compressed geometric primitive or motif. As $M$ grows too large, this compression diminishes, and the bases approach redundant high-dimensional templates.
>
> (iii) **Empirical saturation.** As shown in Figure 7, performance saturates once $K$ and $M$ reach modest values. To further validate this, we provide a direct stress study on **COLLAB**:
>
> | Setting | Variant | Avg. Time / Epoch (s) | Peak Memory (GB) | 5-shot Acc. (%) |
> | :--- | :---: | :---: | :---: | :---: |
> | **Default** (batch size=128) | $K=16, M=32$ | 5.12 ± 0.66 | 0.23 | 66.40 ± 6.33 |
> | **Scale $K$** (Fix $M=32$) | $K=32$ | 5.77 ± 0.38 | 0.34 | 66.12 ± 6.12 |
> |  | $K=64$ | 5.76 ± 0.35 | 0.61 | 63.83 ± 6.17 |
> | **Scale $M$** (Fix $K=16$) | $M=64$ | 4.85 ± 0.11 | 0.25 | 65.68 ± 5.07 |
> |  | $M=128$ | 4.80 ± 0.12 | 0.37 | 64.53 ± 4.69 |
>
> These results indicate that even increasing the base size to roughly **4×** the default configuration incurs only modest additional memory overhead, while the accuracy gain is negligible or slightly negative. This aligns with the empirical saturation trend, confirming that a compact geometric dictionary is sufficient to capture the structural diversity required for effective transfer.

---

> > ### Author Rebuttal · Reviewer_C5Gu · 2026-04-02
> >
> > The authors‘responses’ have addressed my comments. I maintain my positive score.

---

> > > ### Author Response · Authors · 2026-04-02
> > >
> > > We sincerely thank you for your positive evaluation of our work and for your thoughtful feedback during the rebuttal process. We greatly appreciate your recognition and are committed to carefully incorporating your constructive suggestions into the final version of the paper.

---

### Official Review · Reviewer_criL · 2026-03-13

**Soundness:** 2
**Presentation:** 3
**Significance:** 3
**Originality:** 3
**Overall Recommendation:** 4
**Confidence:** 3

**Summary:**

In this paper, the authors proposed Structure-Centric Graph Foundation Models (SCGFM) to address the limitations of existing GFMs caused by structural heterogeneity and incompatible node feature spaces. The method models graphs as metric measure spaces and introduces learnable geometric bases with Gromov–Wasserstein alignment to produce structure-aligned latent representations. Experiments on graph- and node-level tasks show strong in-domain and cross-domain generalization, outperforming existing GFM approaches.

**Compliance With Llm Reviewing Policy:**

Affirmed.

**Final Justification:**

My concerns have been addressed, I keep score.

**Key Questions For Authors:**

See weaknesses

**Limitations:**

no discussion

**Strengths And Weaknesses:**

Strengths:

1. This paper addresses an important challenge in graph foundation models: structural heterogeneity and feature incompatibility across graph domains.

2. The paper provides theoretical analysis to support the proposed method.

3. Experimental results suggest strong performance in cross-domain and few-shot settings across multiple datasets.

Weaknesses:

1. The proposed method relies on Gromov–Wasserstein (GW) alignment, which is known to be computationally expensive, especially for large graphs. Although the paper includes experiments on several datasets, the computational cost of the alignment step is not thoroughly analyzed. In particular, the paper would benefit from: A time and memory complexity analysis of the SCGFM pipeline, runtime comparisons with baselines to demonstrate practical scalability.

2. Experimental comparisons with stronger or more diverse baselines could strengthen the evaluation.
While the paper compares with several graph foundation model approaches, it is unclear whether the evaluation includes recent graph pretraining and cross-domain graph learning methods that address similar transferability challenges. In particular:
It would be helpful to include comparisons with additional GFM or graph pretraining methods, especially those designed for cross-domain transfer or heterogeneous graphs.

---

> ### Author Rebuttal · Authors · 2026-03-29
>
> **Q1.** *"A time and memory complexity analysis of the SCGFM pipeline."*
>
> Thank you for this suggestion. As discussed on page 7 and in Fig.6, SCGFM demonstrates strong scalability. We provide a more detailed analysis below.
>
> (i) **Training complexity.** The main computational bottleneck lies in metric alignment during pre-training. We adopt **Sliced Gromov-Wasserstein (SGW)**, which reduces the dominant cost to
> $$
> O\bigl(KL(N\log N + M\log M)\bigr),
> $$
> where $N$ and $M$ are the numbers of nodes in the input graph and geometric base, respectively, $K$ is the number of bases, and $L$ is the number of slices. Since $K$, $L$, and $M$ are small parameters independent of graph size, the resulting scaling is **near-linear in practice**.
>
> (ii) **Inference complexity.** The inference stage consists of three parts:
>
> - **Structural coordinate computation** $\mathbf{w}$ via SGW: $O\bigl(KL(N\log N + M\log M)\bigr)$
> - **Nonlinear projection** $f(\mathbf{w})$: a lightweight MLP with negligible cost $O(K^2)$
> - **Feature projection** $\mathbf{H}$ via entropic GW: $O(NM \cdot iter)$, where $iter$ is the maximum iteration number.
>
> Therefore, the total inference complexity is:
> $$
> O\bigl(KL(N\log N + M\log M) + K^2 + NM\cdot iter\bigr).
> $$
>
> (iii) **Memory complexity.** SCGFM stores only a compact set of geometric bases (e.g., $16 \times 20 \times 20$), and memory usage is dominated by the input graph adjacency plus the intermediate sliced projections. As a result, it achieves an overall memory complexity of $O(N + |E|)$.
>
> As further empirical evidence, the scalability experiment in Fig.6 demonstrates that SCGFM remains stable even under an extreme stress test with **5M nodes**, reaching a peak memory usage of approximately **12 GB**. Notably, SCGFM is the **only** method among the compared baselines that successfully completes the 5M-node setting.
>
> Overall, both the theoretical analysis and the stress tests confirm that SCGFM achieves favorable scalability, owing to the SGW approximation and its compact geometric base design.
>
> **Q2.** *"Runtime comparisons with baselines to demonstrate practical scalability."*
>
> As illustrated in Fig. 6, we conducted comparative experiments on synthetic datasets with a fixed batch of 1,000 graphs, while varying the number of nodes per graph. SCGFM demonstrates superior stability as it scales to massive graphs. The **raw data** are provided below.
>
> |Node Number|Average Edges|GAT|GIN|GIT|RiemannGFM|SCGFM|
> |-|-|-|-|-|-|-|
> |0.1M|494.67|0.12s|0.10s|0.21s|0.24s|0.84s|
> |0.5M|12468.16|0.70s|0.19s|1.15s|0.54s|0.89s|
> |1.0M|49954.51|2.61s|0.61s|3.73s|1.46s|1.47s|
> |3.0M|449841.264|OOM|4.81s|OOM|8.46s|7.69s|
> |5.0M|1249773.292|OOM|OOM|OOM|OOM|**19.96s**|
>
> These results indicate that SCGFM incurs a slightly higher constant overhead on small graphs due to metric alignment. However, its runtime scales much more gracefully with increasing graph size, and it is the **only** method that remains executable on 5.0M nodes within a 24 GB VRAM budget. In contrast, all other advanced baselines encounter out-of-memory failures before reaching this scale.
>
> **Q3.** *"It would be helpful to include comparisons with additional GFM or graph pretraining methods, especially those designed for cross-domain transfer or heterogeneous graphs."*
>
> While our original submission already provides a comprehensive comparison of SCGFM against 11 representative baselines (including SSL and GFM-style models such as **GIT** and **RiemannGFM**), we thank the reviewer for the valuable suggestion to include more recent methods. In response, we have further strengthened our evaluation by incorporating three additional recent baselines to better validate the cross-domain transferability of SCGFM: **SAMGPT** (WWW 2025), **BRIDGE** (ICML 2025) and **MDGFM** (ICML 2025).
>
> All baselines are evaluated under the same **5-shot** cross-domain graph classification setting. Since their default downstream strategies differ from SCGFM, we present two complementary setups to ensure a fair comparison in heterogeneous scenarios:
>
> - **native**: Retain each baseline’s original target-side prompt or adaptation mechanism to the extent supported by the available code.
> - **aligned**: Freeze the pretrained encoder as much as possible, minimize target-side adaptation, and evaluate using ProtoNet under the same fixed-encoder protocol applied to SCGFM.
>
> The resulting supplementary comparison is as follows:
>
> | Model | BZR -> PROTEINS (%) | PROTEINS -> BZR (%)|
> | ----- | ----------------- | ----------------- |
> | BRIDGE (native) | 51.06 ± 5.80 | 56.38 ± 6.89 |
> | SAMGPT (native) | 51.36 ± 5.52 | 57.90 ± 7.47 |
> |MDGFM (native)|57.06 ± 7.70|53.00 ± 6.02|
> | BRIDGE (aligned) | 49.94 ± 0.42 | 51.24 ± 5.94 |
> | SAMGPT(aligned) | 50.16 ± 0.89 | 53.12 ± 7.08 |
> |MDGFM (aligned)|54.04 ± 6.78|51.94 ± 5.30|
> | SCGFM | **68.32 ± 4.65** | **58.60 ± 6.75** |
>
> SCGFM is best in both aligned directions and also surpasses the strongest native baseline in both transfer directions.

---

> > ### Author Rebuttal · Reviewer_criL · 2026-04-04
> >
> > My concerns have been addressed, I will remain my positive rating.

---

### Decision · Program_Chairs · 2026-04-30

**Decision:**

Accept (regular)

**Comment:**

This paper proposed Structure-Centric Graph Foundation Models (SCGFM) to address the limitations of existing GFMs caused by structural heterogeneity and incompatible node feature spaces. All reviewers agree to accept it.